# E3 ligase FBXW7 is critical for RIG-I stabilization during antiviral responses

Yinjing Song[1,*], Lihua Lai[1,*], Zhenlu Chong[1,*], Jia He[1], Yuanyuan Zhang[2], Yue Xue[1], Yiwei Xie[1], Songchang Chen[1], Ping Dong[3], Luoquan Chen[1], Zhimin Chen[2], Feng Dai[1], Xiaopeng Wan[1], Peng Xiao[1], Xuetao Cao[1,4], Yang Liu[1] & Qingqing Wang[1]

Viruses can escape from host recognition by degradation of RIG-I or interference with the RIG-I signalling to establish persistent infections. However, the mechanisms by which host cells stabilize RIG-I protein for avoiding its degradation are largely unknown. We report here that, upon virus infection, the E3 ubiquitin ligase FBXW7 translocates from the nucleus into the cytoplasm and stabilizes RIG-I. FBXW7 interacts with SHP2 and mediates the degradation and ubiquitination of SHP2, thus disrupting the SHP2/c-Cbl complex, which mediates RIG-I degradation. When infected with VSV or influenza A virus, FBXW7 conditional knockout mice (Lysm$^+$FBXW7$^{f/f}$) show impaired antiviral immunity. FBXW7-deficient macrophages have decreased RIG-I protein levels and type-I interferon signalling. Furthermore, PBMCs from RSV-infected children have reduced FBXW7 mRNA levels. Our results identify FBXW7 as an important interacting partner for RIG-I. These findings provide insights into the function of FBXW7 in antiviral immunity and its related clinical significance.

[1] Institute of Immunology, Zhejiang University School of Medicine, 866 Yuhangtang Road, Hangzhou, Zhejiang 310058, China. [2] The Children's Hospital, Zhejiang University School of Medicine, Hangzhou 310006, China. [3] Department of Neurobiology, Zhejiang University School of Medicine, Hangzhou 310058, China. [4] National Key Laboratory of Medical Molecular Biology and Department of Immunology, Chinese Academy of Medical Sciences, Beijing 100005, China. * These authors contributed equally to this work. Correspondence and requests for materials should be addressed to Q.W. (email: wqq@zju.edu.cn) or to Y.L. (email: liuyang0620@zju.edu.cn).

As the frontier of host defense, the innate immune system detects and fights against viruses, which depend on the detection of pathogen-associated molecular patterns by several classes of germline-encoded pattern-recognition receptors, including Toll-like receptors (TLRs), retinoic acid-inducible gene I (RIG-I)-like receptors (RLRS), Nod-like receptors and DNA sensors[1,2]. TLR3, 7/8 and 9 recognize virus-derived double-stranded RNA, single-stranded RNA and unmethylated CpG DNA, respectively, in the endosome, leading to the activation of nuclear factor-κB and interferon (IFN) regulatory transcription factor (IRF)-3/7 (ref. 1). The cGAS, IFI16 and DDX41 detect double-stranded DNA in the cytosol and adaptor protein STING is mobilized to activate the type-I IFN response[3–5]. In parallel, the RLR member RIG-I and melanoma differentiation-associated gene 5 (MDA5) recognize viral RNAs in the cytoplasm and activate the mitochondrial signalling adaptor (MAVS) to trigger the production of type-I IFNs and proinflammatory cytokines[6]. The type-I IFN is critical in the innate immune response against viral infections[7], which induces a large number of multifunctional IFN-stimulated genes (ISGs)[8]. MDA5 specifically recognizes picornaviruses such as encephalomyocarditis virus (EMCV)[9]. RIG-I, the most important member of RLRs, recognizes a wide variety of RNA viruses such as respiratory syncytial virus (RSV), hepatitis B and C virus, and influenza virus[10–14]. It has been reported that viruses such as HCV can develop strategies to escape from the host recognition through a complex process combination including RIG-I signalling interference, degradation or inhibition of RIG-I expression, or inhibition of effector modulation to support virus replication and persistence[12,15,16]. Defining the molecular mechanisms by which viruses regulate the host response and how host cells regulate the RIG-I response are of crucial importance and may reveal targets for therapeutic strategies.

As one of the most versatile posttranslational modifications, ubiquitination plays a critical role in regulating type-I IFN signalling. The E3 ligase tripartite motif containing protein 25 (ref. 17), Riplet (also known as RNF135)[18], tripartite motif containing protein 4 (ref. 19) and MEX3C[20] have been shown to conjugate covalent K63-polyubiquitin chains to RIG-I at the carboxy terminus to activate RIG-I and promote a massive aggregation of MAVS, which activate the transcription factor IRFs and nuclear factor-κB. The deubiquitinating enzyme ubiquitin-specific protease 4 was identified as a new regulator for RIG-I activation through deubiquitination and stabilization of RIG-I[21]. On the other hand, it has been reported that RIG-I could be degraded by ubiquitination. The RING E3 ligase RNF125 (ref. 22) was also reported to negatively regulate the RLR signalling pathway by binding and ubiquitinating RIG-I with K48-linked polyubiquitin chains, leading to RIG-I degradation through ubiquitin–proteasome system. Our previous study has identified a Siglec-G-mediated immune evasion pathway exploited by RNA viruses through promoting RIG-I degradation and we revealed that recruitment of SHP2 and the E3 ubiquitin ligase c-Cbl to RIG-I leads to RIG-I degradation via K48-linked ubiquitination[23]. Thus, the maintenance of RIG-I protein stability is critical for efficient type-I IFN production and initiation of antiviral immunity. However, the mechanisms how host cells stabilize RIG-I protein for avoiding its degradation are largely unknown.

FBXW7 (F-box and WD repeat domain-containing 7) is a component of SCF (complex of SKP1, CUL1 and F-box protein)-type ubiquitin ligase, which is responsible for recognizing and binding substrates[24]. The SCF[FBXW7] complex promotes the ubiquitination and subsequent degradation of multiple proteins including cyclin E[25], c-Myc[26], c-Jun[27], Notch[28], MCL1 (ref. 29), P100 (ref. 30) and KLF5 (ref. 31). Thus, it plays central roles in tumorigenesis, lipid metabolism, cell proliferation, stemness and differentiation[32]. However, the function of FBXW7 in antiviral immunity has not been reported.

Here a decrease of FBXW7 messenger RNA expression was detected in the peripheral blood mononuclear cells (PBMCs) from 70 cases of children infected with RSV when compared with PBMCs from control healthy children. We generated LysM-cre FBXW7[f/f] conditional knockout (KO) mice that specifically targets deletion in the myelomonocytic and osteoclast lineages. Interestingly, we found that FBXW7 positively regulates antiviral innate immunity. Lysm[+] FBXW7[f/f] mice are more sensitive to RSV, Influenza A (H1N1) virus and vesicular stomatitis virus (VSV) infection and produced less IFN-β and IFN-α4 in macrophages. Surprisingly, we found that, upon VSV infection, FBXW7 translocates from the nucleus into the cytoplasm, to stabilize RIG-I by disrupting the SHP2/c-Cbl complex, which mediates RIG-I degradation, thus promoting downstream signalling. Our data demonstrates that FBXW7 plays an important role in antiviral immune responses by maintaining the stability of RIG-I.

## Results

**FBXW7-deficient mice are more sensitive to RNA virus.** To investigate whether FBXW7 played a possible role in human defense against viral infection, we collected the peripheral blood samples from 70 cases of paediatric patients infected with RSV, who admitted to Children's Hospital, Zhejiang University School of Medicine, and 40 healthy children as a control group. We detected the FBXW7 mRNA expression in the PBMCs and found that FBXW7 mRNA expression in PBMCs of patients decreased in comparison with that in the healthy group. Among these patients, the FBXW7 mRNA was significantly lower in PBMCs from 20 cases of moderate infection in comparison with the 40 cases of mild infection (Fig. 1a), indicating that the expression of FBXW7 in PBMCs might have a correlation with the antiviral immunity of the host. To investigate the role of FBXW7 in host antiviral immunity, we silenced FBXW7 with small interfering RNA (siRNA) in THP-1 cells (Supplementary Fig. 1a) and found the IFN-β mRNA expression was significantly attenuated after being infected with RSV (Fig. 1b), whereas the RSV-G mRNA was significantly increased (Fig. 1c). In addition, the tumour necrosis factor (TNF)-α and interleukin (IL)-6 mRNA expression was downregulated in FBXW7-silenced THP1 cells infected with RSV (Supplementary Fig. 1b). We found the same phenomena in FBXW7-silenced THP1 cells infected with VSV (Fig. 1d,e). To further confirm the function of FBXW7 in innate immunity, we generated FBXW7 conditional KO mice that specifically targets deletion in the myelomonocytic and osteoclast lineages by crossing FBXW7[f/f] mice with LysM-Cre transgenic mice and confirmed the successful loss of F-box domain of FBXW7, which is essential for SCF assembly in FBXW7[f/f] mouse macrophages and dendritic cells (DCs; Supplementary Fig. 1c–e). The proportion of macrophages, DCs and granulocytes in the spleen was comparable between FBXW7[f/f] littermates and Lsm[+] FyBXW7[f/f] mice (Supplementary Fig. 1f-g). Thus, the deletion of FBXW7 in myeloid lineage does not affect the development of myeloid cell subsets.

We then investigated the function of FBXW7 in host defense against RNA virus infection in vivo. Viruses ($8 \times 10^7$ p.f.u. g$^{-1}$) were injected via the tail vein, Lysm[+] FBXW7[f/f] mice displayed more sensitive to VSV infection in overall survival assay (Fig. 1f). In addition, we infected mice with H1N1 virus by intranasal inoculation and Lysm[+] FBXW7[f/f] mice showed higher sensitivity to H1N1 virus infection (Supplementary Fig. 2a). The lung tissue damage was more severe along with more inflammatory cell

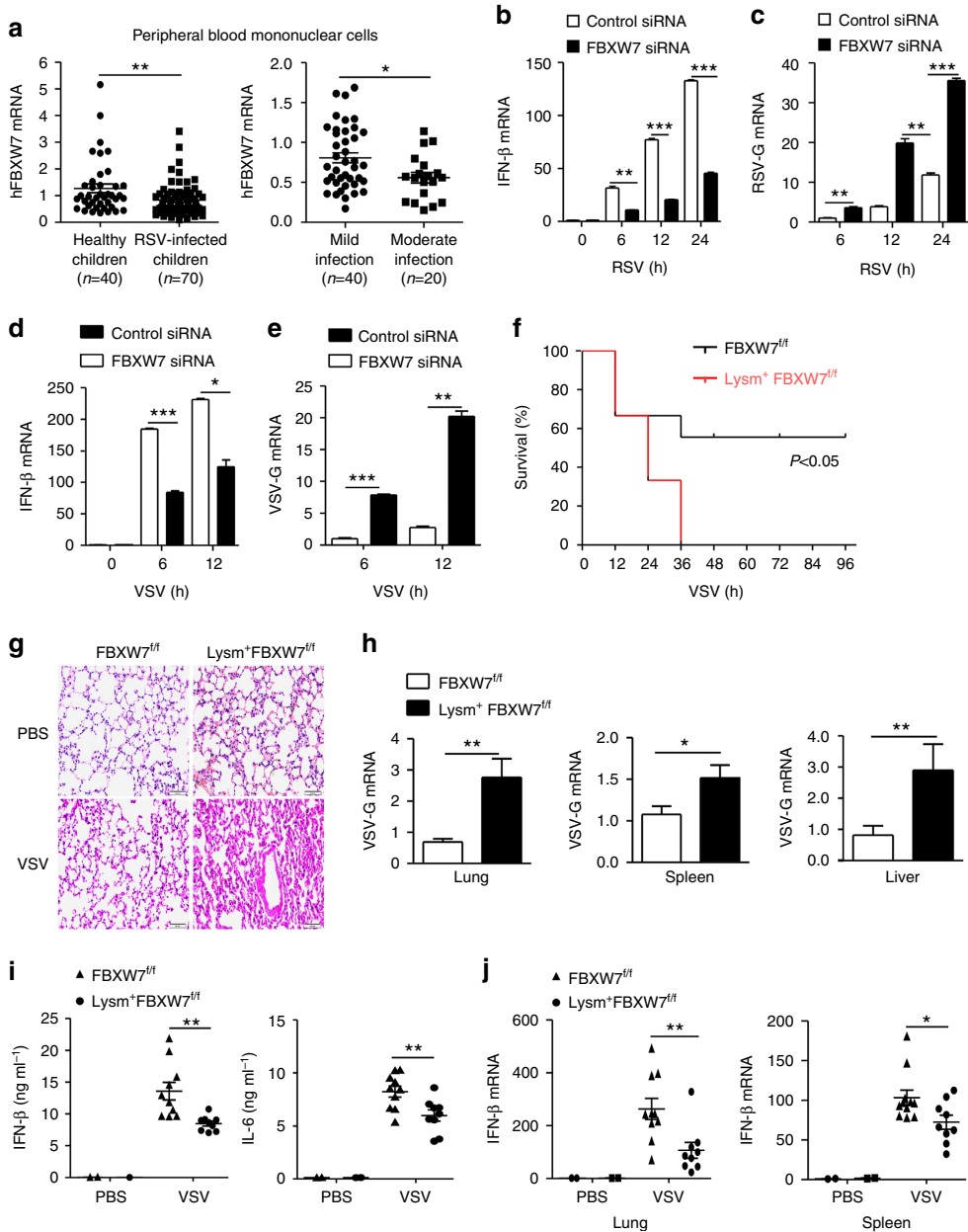

**Figure 1 | FBXW7-deficient mice are more sensitive to RNA virus infection by producing more type I IFN.** (**a**) Quantitative PCR (Q-PCR) analysis of FBXW7 mRNA expression in PBMCs of 70 cases of paediatric patients infected with RSV and 40 healthy children (left), and compared the FBXW7 expression in PBMCs between mild infection (40 cases) and moderate infection (20 cases) (right). Q-PCR analysis of IFN-β (**b**) and VSV-G mRNA (**c**) expression in THP-1 cells transfected for 36 h with siFBXW7 and infected with RSV. Q-PCR analysis of IFN-β (**d**) and VSV-G mRNA (**e**) expression in THP-1 cells transfected for 36 h with si-FBXW7 and infected with VSV. (**f**) Survival assay of ∼7-week-old FBXW7$^{f/f}$ and Lysm$^+$FBXW7$^{f/f}$ mice injected with VSV ($8 \times 10^7$ p.f.u. g$^{-1}$) via the tail vein. $n = 9$ per group. Kaplan–Meier survival curves were generated and analysed. (**g**) Lung histology of FBXW7$^{f/f}$ and Lysm$^+$FBXW7$^{f/f}$ mice in response to VSV. Scale bar, 50 μm. Haematoxylin and eosin staining of lung tissues from mice infected with VSV ($1 \times 10^8$ p.f.u. g$^{-1}$) by intraperitoneal injection. (**h**) Q-PCR analysis of VSV-G expression in organs from FBXW7$^{f/f}$ and Lysm$^+$FBXW7$^{f/f}$ mice in **g**. (**i**) ELISA assay of cytokines in sera from mice as in **g**. (**j**) Q-PCR analysis of IFN-β expression in the spleen and lung from mice in **g**. Data are mean ± s.e.m. and are representative of three independent experiments. Student's *t*-test was used for statistical calculation. *$P < 0.05$, **$P < 0.01$ and ***$P < 0.001$.

infiltration in Lysm$^+$FBXW7$^{f/f}$ mice compared with FBXW7$^{f/f}$ mice after VSV (Fig. 1g) or H1N1 virus infection (Supplementary Fig. 2b). Moreover, the replication of VSV (Fig. 1h) or H1N1 virus (Supplementary Fig. 2c) was significantly increased in organs from KO mice. Lysm$^+$FBXW7$^{f/f}$ mice produced significantly lower levels of IFN-β and IL-6 than FBXW7$^{f/f}$ mice in response to VSV (Fig. 1i) or H1N1 virus (Supplementary Fig. 2d). In addition, we detected decreased IFN-β mRNA expression in the spleen and lung from Lysm$^+$FBXW7$^{f/f}$ mice

compared with those from FBXW7$^{f/f}$ mice infected with VSV (Fig. 1j) or H1N1 virus (Supplementary Fig. 2e). However, no significant difference in IFN-β production was observed between FBXW7 wild-type (WT) and KO mice challenged with DNA virus, herpes simplex virus-1 (Supplementary Fig. 2f). It implied that the upregulation of IFN-β by FBXW7 is specific to RNA virus infection. These results indicated a function of FBXW7 in protecting the host against RNA virus infection by promoting the production of type-I IFN *in vivo*.

**FBXW7 inhibits RNA virus infection *in vitro*.** The FBXW7-deficient peritoneal macrophages expressed downregulated IFN-β and IFN-α4 mRNA upon infection with VSV, H1N1 virus or RSV (Fig. 2a) and this phenomenon was also found in bone marrow-derived DCs (BMDCs; Fig. 2b) and bone marrow-derived macrophages (BMDMs; Supplementary Fig. 3a). Enzyme-linked immunosorbent assay (ELISA) assay showed that deletion of FBXW7 in macrophages and BMDC resulted in downregulated

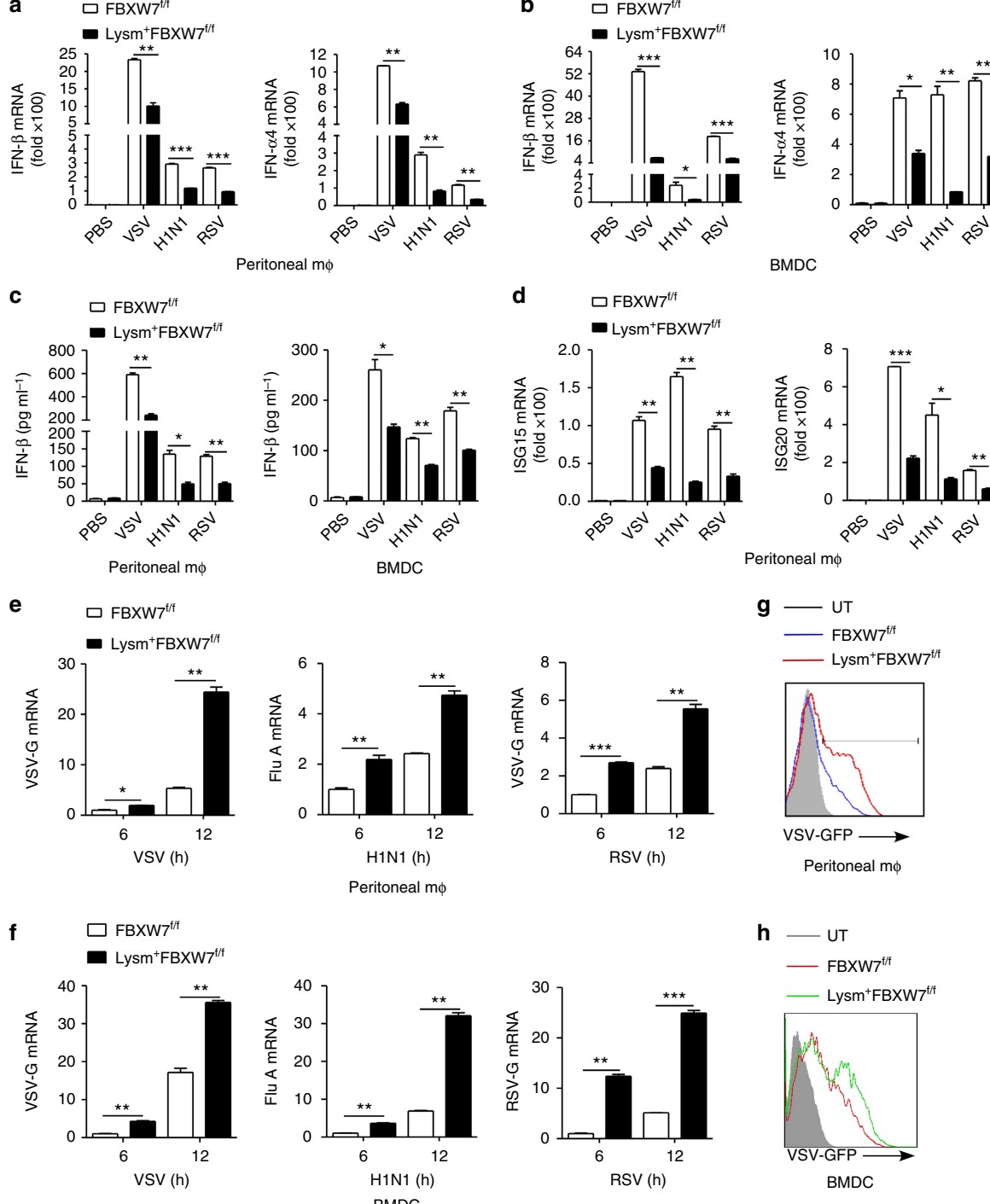

**Figure 2 | FBXW7 inhibits RNA virus infection and promotes type-I IFN production *in vitro*.** (**a–b**) Quantitative PCR (Q-PCR) analysis of IFN-β, IFN-α4 mRNA expression in FBXW7$^{f/f}$ and Lysm$^{+}$FBXW7$^{f/f}$ peritoneal macrophages (**a**) or BMDC (**b**) infected with VSV, H1N1 virus or RSV. (**c**) ELISA assay of IFN-β in supernatants of FBXW7$^{f/f}$ and Lysm$^{+}$FBXW7$^{f/f}$ peritoneal macrophages or BMDC infected for 24 h with VSV, H1N1 virus or RSV. (**d**) Q-PCR analysis of ISG15 and ISG20 mRNA expression in FBXW7$^{f/f}$ and Lysm$^{+}$FBXW7$^{f/f}$ peritoneal macrophages infected with VSV, H1N1 virus or RSV. (**e–f**) Q-PCR analysis of VSV-G, Flu A and RSV-G mRNA expression in FBXW7$^{f/f}$ and Lysm$^{+}$FBXW7$^{f/f}$ peritoneal macrophages (**e**) or BMDC (**f**) infected with VSV, H1N1 or RSV for 12 h. (**g,h**) Flow cytometry analysis of GFP fluorescence intensity in FBXW7$^{f/f}$ and Lysm$^{+}$FBXW7$^{f/f}$ peritoneal macrophages (**g**) or BMDC (**h**) infected with VSV-GFP. Data are mean ± s.e.m. and are representative of three independent experiments. Student's *t*-test was used for statistical calculation. *$P < 0.05$, **$P < 0.01$ and ***$P < 0.001$.

IFN-β production (Fig. 2c), when cells were infected with VSV, H1N1 virus or RSV. Consistently, deletion of FBXW7 in macrophages also led to decreased expression of ISG15 and ISG20 mRNAs after infection with VSV, H1N1 virus or RSV (Fig. 2d). It is well known that short poly (I:C) selectively activates RIG-I, whereas long poly (I:C) induces IFN signalling through MDA5. Of the low-molecular-weight (LMW) poly (I:C), with an average of $0.2\sim1$ kb, specifically recognized by RIG-I, and high-molecular-weight poly (I:C), with an average of $1.5\sim8$ kb, recognized both by RIG-I and MDA5 (ref. 33), we chose LMW poly (I:C) to transfect peritoneal macrophages, to trigger the IFN signalling. Lysm$^+$FBXW7$^{f/f}$ macrophages expressed much lower IFN-β and IFN-α4 mRNA level than FBXW7$^{f/f}$ macrophages in response to intracellular transfection of LMW-poly (I:C) (Supplementary Fig. 3b). The replication of VSV, H1N1 virus or RSV was significantly increased in FBXW7-deficient peritoneal macrophages (Fig. 2e) and BMDCs (Fig. 2f). In addition, the FBXW7-deficient macrophages (Fig. 2g) and BMDCs (Fig. 2h) showed increased GFP$^+$ cells when infected with VSV-GFP.

To further confirm the function of FBXW7 in vitro, we silenced FBXW7 in peritoneal macrophages with siRNA (Supplementary Fig. 3c) and detected the decreased expression of IFN-α4, IFN-β mRNA in FBXW7-silenced peritoneal macrophages upon VSV or H1N1 virus infection (Supplementary Fig. 3d). In addition, the replication of VSV was dramatically increased in FBXW7-silenced macrophages (Supplementary Fig. 3e,f), whereas overexpression of FBXW7 induced elevated IFN-β mRNA in response to VSV (Supplementary Fig. 3g). The cells with FBXW7 overexpression showed fewer GFP$^+$ cells compared with cells transfected with control plasmid (Supplementary Fig. 3h) and less VSV-G mRNA level (Supplementary Fig. 3i) in response to VSV. These data suggested that FBXW7 promoted type-I IFN and enhanced antiviral response to RNA virus in vitro.

**FBXW7 promotes RIG-I signalling by protecting RIG-I from degradation.** Next, we investigated the effect of FBXW7 on RNA virus downstream signal pathways in macrophages. The results showed that the phosphorylation of IRF3 and TBK1 was significantly inhibited in Lysm$^+$FBXW7$^{f/f}$ macrophages relative to FBXW7$^{f/f}$ macrophages infected with VSV (Fig. 3a). Meanwhile, p65, p38 and ERK phosphorylations were also downregulated in Lysm$^+$FBXW7$^{f/f}$ macrophages (Fig. 3a). We observed similar phenomenon in macrophages transfected with LMW poly (I:C) (Supplementary Fig. 4a). FBXW7-deficient macrophages showed lower RIG-I protein level upon VSV infection (Fig. 3b) or LMW poly (I:C) transfection (Supplementary Fig. 4b). However, no significant difference in TBK1 and MAVS protein level was detected between FBXW7 WT and KO macrophages (Fig. 3b and Supplementary Fig. 4c). Considering the previous report that FBXW7 locates in the nucleus[24], we speculated that FBXW7 might regulate the mRNA level of RIG-I. We compared the RIG-I mRNA level in WT and KO macrophages infected with VSV. Surprisingly, there was no difference in RIG-I mRNA level between FBXW7 KO and WT macrophages (Fig. 3c). The cycloheximide chase assay of macrophages showed that FBXW7 deficiency decreased the half-life of endogenous RIG-I protein during VSV infection (Fig. 3d,e). These data indicated that FBXW7 protects RIG-I from degradation and FBXW7 might regulate RIG-I in a posttranscription manner. To address this question, we utilized immunofluorescent assay to observe the interaction between FBXW7 and RIG-I along with VSV infection. Interestingly, we found that FBXW7 co-localizes with RIG-I (Fig. 3f) during response to VSV infection. Furthermore, we used coimmunoprecipitation and immunoblotting to confirm the interaction between FBXW7 and RIG-I (Fig. 3g). MDA5 is

also an important member of RLRs. To investigate whether FBXW7 also affect MDA5's stability, we detected the protein level of MDA5 between FBXW7$^{f/f}$ and Lysm$^+$FBXW7$^{f/f}$ macrophages during EMCV infection (Supplementary Fig. 4d). There was no significant difference of protein level of MDA5 between FBXW7 WT and KO macrophages. The same phenomenon was observed in macrophages transfected with high-molecular-weight poly (I:C), which could be recognized by MDA5 (Supplementary Fig. 4e). The half-life of MDA5 protein showed no difference between FBXW7 WT and KO macrophages treated with cycloheximide during EMCV infection (Supplementary Fig. 4f,g). The FBXW7-deficient macrophages expressed almost the same levels of IFN-β and IFN-α4 mRNA compared with FBXW7 WT macrophages upon infection with EMCV (Supplementary Fig. 4h). These data indicated that FBXW7 specifically interacted with RIG-I and protected RIG-I from degradation, thus promoting RIG-I signalling pathway.

**The translocation of FBXW7 is critical for RIG-I stabilization.** Previous study reported that only α-isoform of FBXW7 exists in mouse immune organs and FBXW7 is located in the nucleoplasm[24]. However, we found FBXW7 could also be detected in the cytoplasm once HEK293T cells were infected with VSV (Fig. 3f). To confirm FBXW7 could translocate from the nucleus into the cytoplasm during virus infection, we utilized immunofluorescent assay to observe the interaction of FBXW7 and chromosomal maintenance 1 (CRM1, also called Exportin 1), which mediated the nucleo-cytoplasmic transport of cargo proteins[34]. The co-localization of FBXW7 and CRM1 was observed in BMDM infected with VSV and the translocation of FBXW7 was increased along with the VSV infection (Fig. 4a). The interaction between FBXW7 and CRM1 was also detected in FBXW7-overexpressed HEK293T cells during VSV infection (Supplementary Fig. 5a). Western blotting showed FBXW7 was downregulated in the nucleus, but was upregulated in the cytoplasm of HEK293T cells (Fig. 4b). When FBXW7-overexpressed HEK293T cells were treated with Leptomycin B, which binds to the active site (CRM1 cysteine residue 528) and prevents CRM1 binding to the cargo protein, it blocked the translocation of FBXW7 from the nucleus to the cytoplasm (Supplementary Fig. 5b). Nuclear protein export is mediated by nuclear export signal or sequence (NES)[35], which is a short amino acid sequence of four hydrophobic residues ($\Phi$-$x_{2\text{-}3}$-$\Phi$-$x_{2\text{-}3}$-$\Phi$-$x$-$\Phi$, '$\Phi$' is a hydrophobic residue and '$x$' is any other amino acid) and is recognized and bound by exportins such as CRM1. It was reported that F-box proteins (such as FBXO7) have the NES, which was located in the F-box domain, and FBXW7 was also predicted containing NES[36]. The FBXW7 conditional KO mice were construed by deletion of F-box domain of FBXW7 in myelomonocytic lineage, we then observed the location of FBXW7 in FBXW7-deficient BMDM. Immunofluorescent assay showed the translocation of FBXW7 into the cytoplasm was not observed during VSV infection (Supplementary Fig. 5c). In addition, we constructed the mutation plasmids of FBXW7 in which isoleucine at position 284 and leucine at positions 287, 291 and 293 were replaced with glycine and the F-box domain was deleted (Supplementary Fig. 5d). The translocation of FBXW7 from the nucleus into the cytoplasm was blocked in FBXW7-mutated cells during VSV infection (Fig. 4c). These data indicated FBXW7 was able to be exported from the nucleus into the cytoplasm by the exportin CRM1 in response to VSV infection.

To confirm the relationship between the antiviral function of FBXW7 and the translocation of FBXW7, we transfected the WT and mutation plasmids into the cells and measured the expression

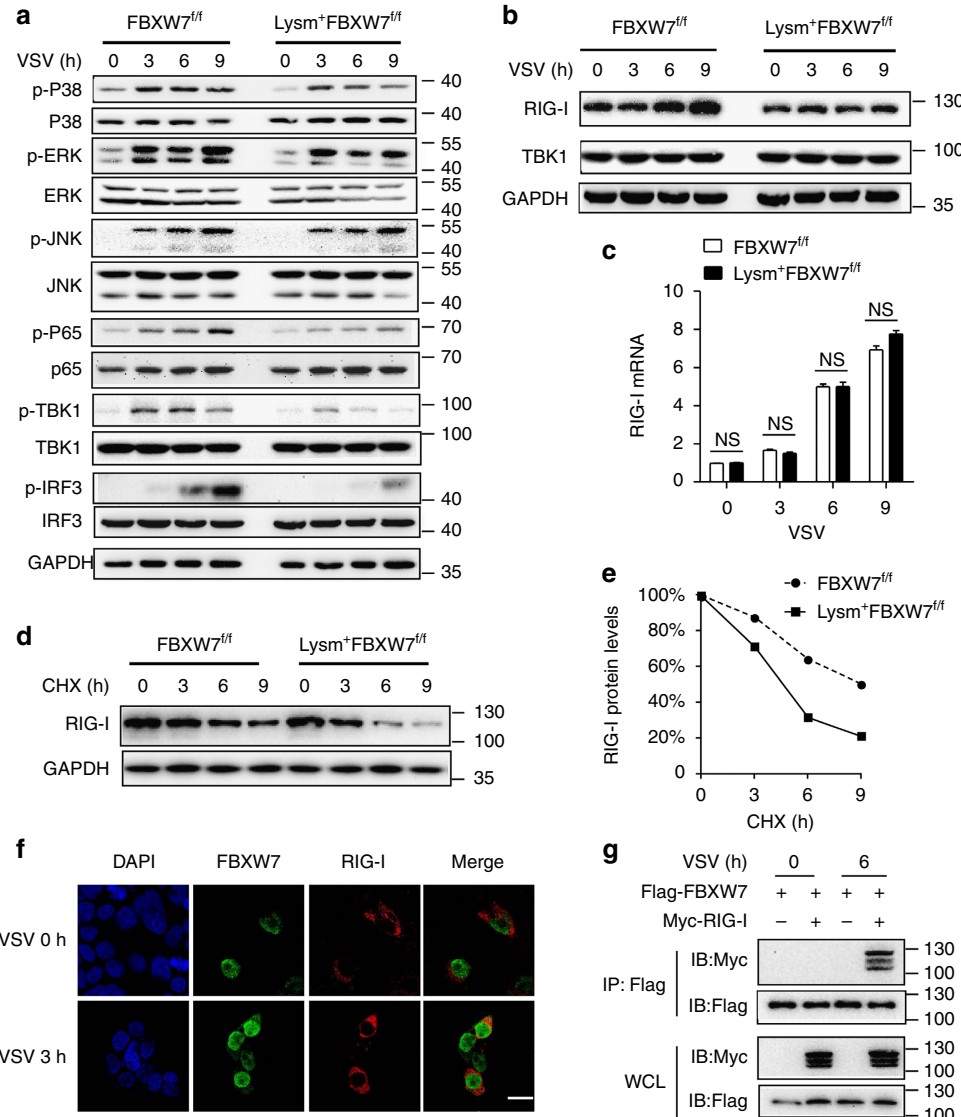

**Figure 3 | FBXW7 positively regulates RIG-I signal pathway through protecting RIG-I from degradation.** (**a**) Immunoblot analysis of phosphorylated or total proteins in lysates of FBXW7$^{f/f}$ and Lysm$^+$FBXW7$^{f/f}$ peritoneal macrophages infected for indicated hours with VSV. (**b**) Immunoblot of RIG-I and TBK1 protein levels in lysates of FBXW7$^{f/f}$ and Lysm$^+$FBXW7$^{f/f}$ macrophages infected with VSV. (**c**) Quantitative PCR (Q-PCR) analysis of RIG-I mRNA expression in FBXW7$^{f/f}$ and Lysm$^+$FBXW7$^{f/f}$ macrophages infected for indicated hours with VSV. Data are mean ± s.e.m. and are representative of three independent experiments. Student's *t*-test was used for statistical calculation. NS, not significant. (**d**) Immunoblot analysis of RIG-I in lysates of FBXW7$^{f/f}$ and Lysm$^+$FBXW7$^{f/f}$ peritoneal macrophages treated with CHX (40 µg ml$^{-1}$) for indicated hours after infection with VSV for 1 h. (**e**) Quantification of relative RIG-I levels is shown in the right panel. (**f**) Confocal microscopy imaging of HEK293T cells that transfected with Flag-FBXW7, Myc-RIG-I for 24 h and then infected for indicated hours with VSV, and labelled with antibodies to the appropriate protein. Scale bar, 20 µm. (**g**) Coimmunoprecipitation and immunoblot of HEK293T cells transfected for 24 h with Flag-FBXW7 plasmid or Flag-FBXW7 together with Myc-RIG-I followed by VSV infection for indicated hours.

of IFN-β mRNA and VSV-G mRNA. The antiviral function depended on the induced expression of IFN-β by FBXW7 was also abolished in FBXW7-mutated cells (Fig. 4d). FBXW7 (WT)-overexpressed cells showed less GFP$^+$ population compared with cells transfected with control plasmid during GFP-VSV infection (Supplementary Fig. 5e). However, we did not detect the effect of FBXW7 in the cells transfected with mutation plasmids of FBXW7 (Fig. 4d). Immunofluorescent assay (Fig. 4e), immunoprecipitation and immunoblotting (Supplementary Fig. 5f) did not show the interaction between FBXW7 and RIG-I in FBXW7-mutated cells during VSV infection. Overexpression of FBXW7 promoted the upregulation of RIG-I protein level, whereas the mutated protein of FBXW7

did not show this effect (Supplementary Fig. 5g). These results demonstrated that virus-induced translocation of FBXW7 into the cytoplasm is critical for its antiviral function and RIG-I stabilization. It was reported that FBXW7 mainly mediates K48-linked polyubiquitination and subsequent degradation of proteins. However, what made us puzzled is that deletion of FBXW7 resulted in enhanced K48-linked polyubiquitination of RIG-I in macrophages infected with VSV (Fig. 4f) rather than downregulation of K48-linked polyubiquitination, whereas there was no change in K48-linked polyubiquitination of MAVS (Supplementary Fig. 5h). This indicated that RIG-I was not the direct substrate of FBXW7 and made us to further explore the mechanisms involved in the regulation of RIG-I by FBXW7.

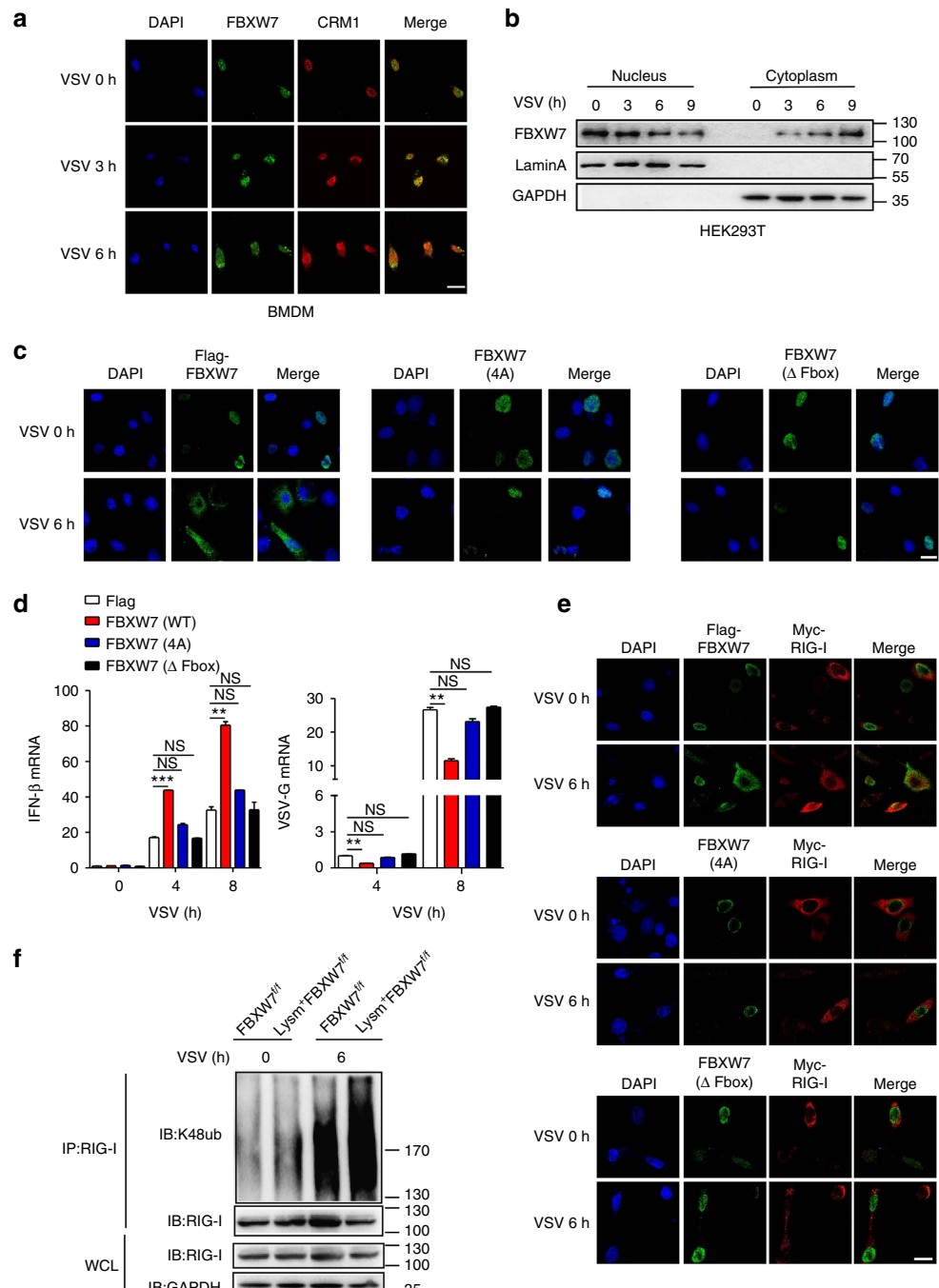

**Figure 4 | FBXW7 translocates into the cytoplasm to interact with RIG-I and exert the antiviral function.** (**a**) Confocal microscopy imaging of BMDM infected for indicated hours with VSV and labelled with antibodies to the appropriate protein. Scale bar, 20 μm. (**b**) Immunoblot analysis of Flag-FBXW7 among nuclear and cytoplasm proteins from overexpressed Flag-FBXW7 HEK293T cells infected with VSV; Lamin A serves as a nuclear control; GAPDH serves as a cytoplasm control. (**c**) Confocal microscopy imaging of HEK293T cells transfected with Flag-FBXW7 (wt) or Flag-FBXW7 (mutant) plasmid for 24 h, then infected for indicated hours with VSV and labelled with antibodies to the appropriate protein. Scale bar, 20 μm. (**d**) Quantitative PCR (Q-PCR) analysis of IFN-β and VSV-G mRNA expression in HEK293T cells transfected with Flag-FBXW7 (wt) or Flag-FBXW7 (mutant) plasmid and infected with VSV for indicated hours. (**e**) Confocal microscopy imaging of HEK293T cells that transfected with Flag-FBXW7 (wt) or Flag-FBXW7 (mutant) together with Myc-RIG-I plasmids followed by VSV infection for 6 h. Scale bar, 20 μm. (**f**) Immunoblot analysis of the K48 ubiquitination of RIG-I in FBXW7^f/f and Lysm+FBXW7^f/f peritoneal macrophages infected with VSV. Data are mean ± s.e.m. and are representative of three independent experiments. Student's *t*-test was used for statistical calculation. **P < 0.01 and ***P < 0.001.

**FBXW7 interacts with SHP2.** To clarify the underlying mechanisms that FBXW7 protecting RIG-I from degradation, FBXW7 was immunoprecipitated from lysates of VSV-infected cells transfected with Flag-FBXW7 by the antibody to Flag epitope tag and mass spectrometry was used to identify FBXW7-interacting proteins. Among the FBXW7-interacting proteins, we focused on the tyrosine–protein phosphatase non-receptor type 11 (SHP2), and SHP2 had been reported to promote RIG-I degradation by recruiting E3 ligase c-Cbl in our previous study[23]. In addition, outstanding mascot scores also

indicated SHP2 to be a candidate for FBXW7-interacting partner (Supplementary Fig. 6a). To confirm this hypothesis, we immunoprecipitated Flag epitope tag from lysates of VSV-infected cells with overexpressed Flag-FBXW7 and immunoblotting revealed that the interaction between FBXW7 and SHP2 was increased by VSV infection (Fig. 5a). Further coimmunoprecipation and immunoblot assay showed that FBXW7 interacted with SHP2 when HEK293T cells transfected with plasmids that express FBXW7 and SHP2 (Fig. 5b). In addition, the endogenous interaction between FBXW7 and SHP2 was also augmented in RAW264.7 cells after VSV infection (Fig. 5c). Confocal microscopy revealed significantly increased co-localization of FBXW7 and SHP2 after VSV infection in peritoneal macrophages (Fig. 5d) and HEK293T cells (Supplementary Fig. 6b).

FBXW7 contains a stretch of eight WD40 repeat domain that recognizes and binds the substrates[37]. We expressed Myc-tagged SHP2 together with Flag-tagged full-length FBXW7 or WD40 domain of FBXW7. Both full-length FBXW7 and the WD40 domain of FBXW7 could bind SHP2 (Fig. 5e). To determine which domain of SHP2 was required for the interaction with FBXW7, we constructed mutants of SHP2 with the deletion of various domains, C-terminal domain of SHP2 was demonstrated to interact with FBXW7 (Fig. 5f). According to previous reports, most of the substrates of FBXW7 contain a conserved phosphorylation site called the CDC4 phosphodegron[38,39]. By analysing C-terminal domain of SHP2, a domain that contained one similar site of the FBXW7 CDC4 phosphodegron, was identified (Fig. 5g). To determine whether this amino acid sequence was required for FBXW7 to recognition and ubiquitination, we constructed mutant plasmid of SHP2 in which serine at position 558 and 562 were replaced with glycine and the result showed that FBXW7 could not bind the mutant SHP2 (Fig. 5h). These results strongly suggested that the serine at position 558 and 562 of SHP2 were critical for its interaction with FBXW7.

**FBXW7 mediates the degradation and ubiquitination of SHP2**. To confirm whether SHP2 was the target of FBXW7, we challenged macrophages with VSV and found higher SHP2 protein expression in FBXW7 KO macrophages than that in WT macrophages (Fig. 6a), but there was no difference in SHP2 mRNA expression (Supplementary Fig. 7a). The cycloheximide chase assay showed that deletion of FBXW7 extended the half-life of intracellular SHP2 protein in peritoneal macrophages infected with VSV (Fig. 6b). Silence of FBXW7 with shRNA also resulted in higher expression of SHP2 (Supplementary Fig. 7b) and extended the half-life of intracellular SHP2 protein in RAW264.7 cells infected with VSV (Supplementary Fig. 7c,d). Overexpression of FBXW7 led to decreased SHP2 protein when infected with VSV (Fig. 6c). To investigate the role of FBXW7 in regulating SHP2, we transfected HEK293T cells with plasmid expressing Flag-SHP2, Myc-FBXW7 together and observed that FBXW7 inhibited SHP2 protein level (Fig. 6d). However, the reduction of SHP2 induced by FBXW7 was blocked by MG132 (Fig. 6e), which indicated that FBXW7 mediates SHP2 degradation via ubiquitin–proteasome system. However, the degradation of the mutant SHP2 (558A and 562A) could not be detected in cells when transfected with Myc-FBXW7 together with mutant or WT SHP2 (Fig. 6f). Collectively, these data indicated that SHP2 could be degraded by FBXW7 via ubiquitin–proteasome system during VSV infection.

To confirm whether SHP2 undergoes protein ubiquitination mediated by FBXW7, we transfected FBXW7 and SHP2 plasmids into HEK293T cells and found that FBXW7 mediated the ubiquitination of SHP2 in a dose-dependent way (Fig. 6g). To investigate which form of the polyubiquitin chains of SHP2 catalysed by FBXW7, we co-transfected plasmids into cells expressing Ha-tagged ubiquitin, the mutant ubiquitin HA-K48R Ub or HA-K63R Ub together with Myc-FBXW7 and Flag-SHP2, and then detected the polyubiquitination of SHP2. Transfection with HA-K48RUb abolished FBXW7-mediated ubiquitination of SHP2 (Fig. 6h). We further used antibody to K48-linked ubiquitin and found that K48-linked ubiquitination of SHP2 was increased in cells overexpressing FBXW7 (Supplementary Fig. 7e). These data indicated that FBXW7 mediates K48-linked polyubiquitina-tion of SHP2. Deletion of FBXW7 in macrophages resulted in dowregulated K48-linked ubiquitination of SHP2 during viral infection (Fig. 6i). Conversely, overexpression of FBXW7 induced enhanced K48-linked polyubiquitination of SHP2 compared with the cells transfected with empty vector (Supplementary Fig. 7f). As the F-box domain of FBXW7 is critical for activity of the E3 ligase SCF$^{FBXW7}$, it showed that FBXW7 lacking the F-box domain did not enhance K48-linked ubiquitin chains (Supplementary Fig. 7g). The Lys91 of SHP2 ubiquitination was discovered by mass spectrometry[40], the SHP2 (K91R) plasmid was constructed in which the site of 91 lysine residues was replaced with arginine and this substitution in cells almost abolished SHP2 ubiquitination compared with the WT SHP2 or SHP2 (K178R) (Fig. 6j). In addition, the SHP2 (K91R) mutant completely blocked the degradation of SHP2 (Fig. 6k). These data demonstrated that FBXW7 functioned as an E3 ligase to catalyse K48-linked ubiquitination of SHP2 at Lys91 through ubiquitin–proteasome system.

**SHP2 mediates the function of FBXW7 stabilizing RIG-I**. To further confirm the role of interaction of SHP2 with FBXW7 in virus infection, we compared IFN-β expression in macrophages from Lysm$^+$SHP2$^{f/f}$ and SHP2$^{f/f}$ mice that infected with VSV. SHP-2-deficient macrophages displayed higher IFN-β and lower VSV-G mRNA level compared with control WT macrophages (Fig. 7a and Supplementary Fig. 8a). Flag-FBXW7 plasmid was transfected into WT and SHP2-deficient macrophages (Supplementary Fig. 8b), it was observed that overexpression of FBXW7 in SHP2 WT macrophages enhanced IFN-β mRNA expression, but not in SHP2 KO macrophages when infected with VSV (Fig. 7b). When FBXW7$^{f/f}$ and Lysm$^+$FBXW7$^{f/f}$ macrophages were transfected with siRNA targeting SHP2, there was no significant difference in IFN-β mRNA in response to VSV (Fig. 7c).

To further determine whether the role of FBXW7 in stabilizing RIG-I and promoting IFN triggered by RNA virus was mediated by SHP2, FBXW7 was immuno-precipitated from lysates of VSV-infected RAW264.7, to detect the interaction between RIG-I and SHP2. The endogenous FBXW7/RIG-I/SHP2 complex was detected in VSV-infected macrophages, whereas the interaction between FBXW7 and TBK1 was not detected (Fig. 7d). It indicates that FBXW7 regulates RIG-I signalling through SHP2/RIG-I instead of SHP2/TBK1, as the interaction between SHP2 and RIG-I was enhanced and predominant during virus infection (Supplementary Fig. 8c). Furthermore, the interaction between RIG-I and SHP2, c-Cbl was enhanced if FBXW7 was deleted in macrophages upon VSV stimulation (Fig. 7e), which indicates FBXW7 disrupted the interaction between RIG-I and SHP2/c-Cbl, thus protecting RIG-I from degradation. Furthermore, the interaction between RIG-I and FBXW7, or RIG-I and c-Cbl was not observed in VSV-infected SHP KO macrophages, indicating that the interaction between FBXW7 and RIG-I is mediated by SHP2 (Fig. 7f). To confirm the direct interaction among RIG-I, FBXW7 and SHP2, these tag proteins

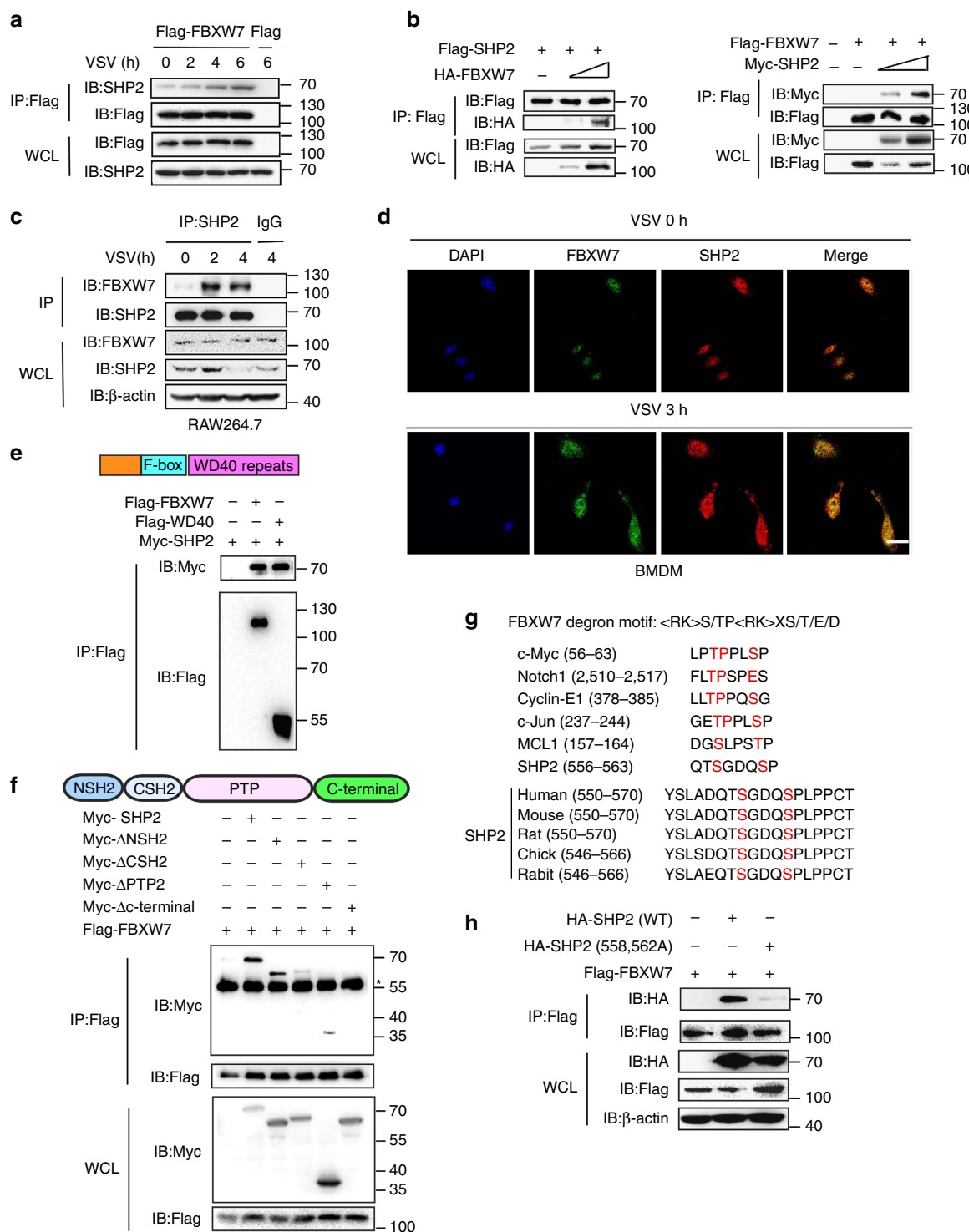

**Figure 5 | FBXW7 interacts with SHP2.** (**a**) Coimmunoprecipitation and immunoblot analysis of HEK293T cells transfected for 24 h with Flag-FBXW7 cells infected with VSV and treated with MG132. (**b**) Coimmunoprecipitation and immunoblot analysis of HEK293T cells cotransfected for 24 h with Flag-SHP2 plus HA-FBXW7 or Flag-FBXW7 and Myc-SHP2, and treated with MG132 followed by immunoprecipitation with anti-Flag-M2 beads. (**c**) Immunoblot analysis of RAW264.7 infected for indicate hours with VSV and treated with MG132, followed by immunoprecipitation with SHP2-conjugated agarose or immunoglobulin G (IgG)-conjugated agarose. (**d**) Confocal microscopy imaging of peritoneal macrophages infected for 3 h with VSV and labelled with antibodies to the appropriate protein. Scale bar, 20 μm. (**e**) Immunoblot analysis of HEK293T cells cotransfected for 24 h with Flag-FBXW7 or Flag-WD40, plus Myc-SHP2 vectors and treated with MG132, followed by immunoprecipitation with anti-Flag-M2 beads. (**f**) Immunoblot analysis of HEK293T cells cotransfected for 48 h with Flag-FBXW7, plus Myc-SHP2, Myc-SHP2 mutants vectors and treated with MG132, followed by immunoprecipitation with anti-Flag-M2 beads. *IgG heavy chain. (**g**) Sequence alignment of SHP2 with FBXW7 degron motif (⟨RK⟩S/TP⟨RK⟩XS/T/E/D), where X is any amino acid and ⟨RK⟩ is any amino acid, except arginine (R) or lysine (K). (**h**) Coimmunoprecipitation and immunoblot analysis of HEK293T cells transfected with Flag-FBXW7 along with vector for HA-SHP2 (WT) and HA-SHP2 (558A and 562A), and treated with MG132, followed by immunoprecipitation with anti-Flag-M2 beads.

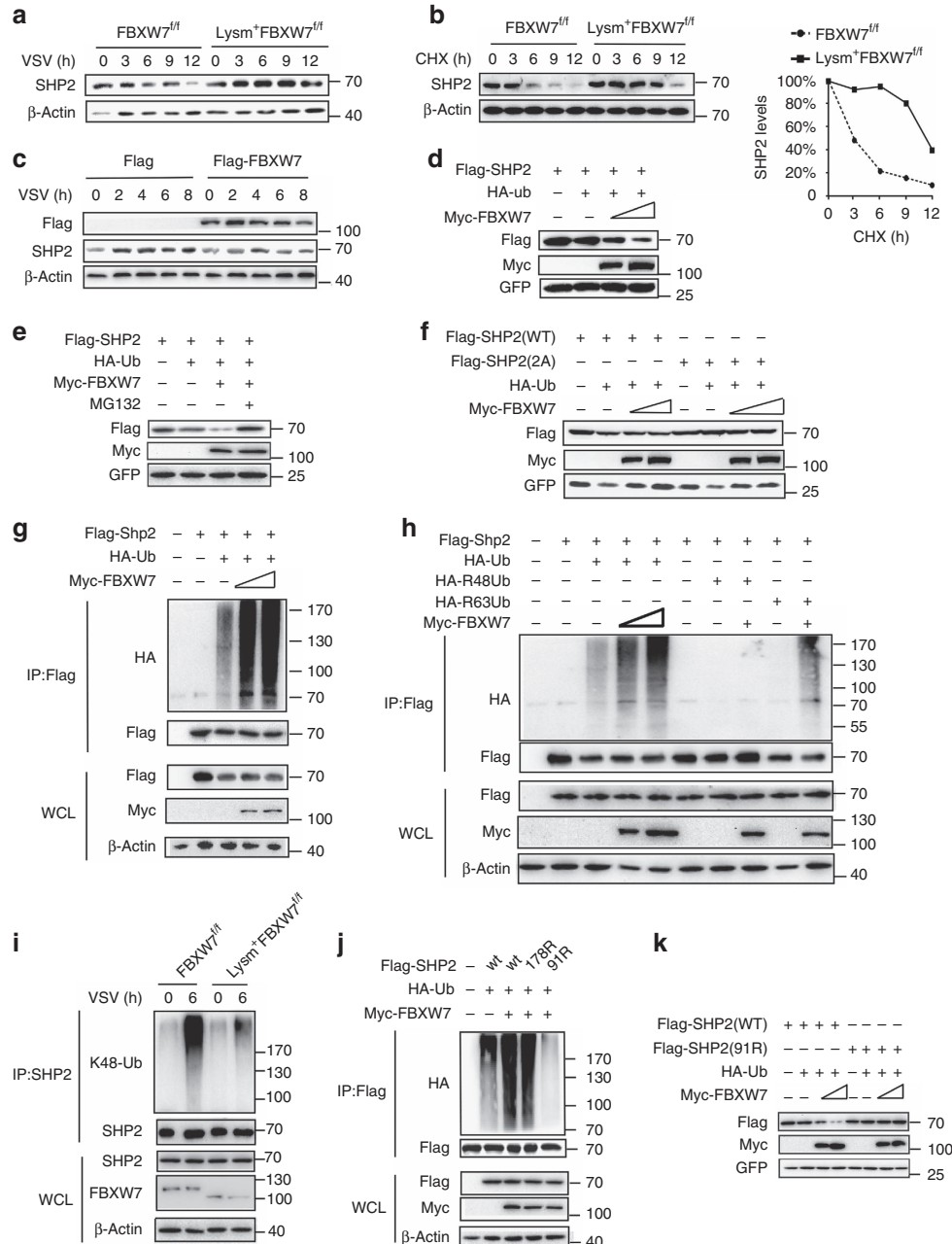

**Figure 6 | FBXW7 mediates the degradation and ubiquitination of SHP2.** (**a**) Immunoblot analysis of SHP2 in lysates of FBXW7[f/f] and Lysm[+]FBXW7[f/f] peritoneal macrophages infected with VSV. (**b**) Immunoblot analysis of SHP2 in lysates of FBXW7[f/f] and Lysm[+]FBXW7[f/f] peritoneal macrophages treated with CHX (40 µg ml[−1]) for indicated hours after infection with VSV for 1 h. Quantification of relative SHP2 levels is shown in the right panel. (**c**) Immunoblot analysis of SHP2 in lysates of HEK293T cells with overexpressed FBXW7 and infected with VSV. (**d**) Immunoblot analysis of HEK293T cells cotransfected for 24 h with Myc-FBXW7, plus Flag-SHP2, HA-ub and GFP vectors. (**e**) Immunoblot analysis of HEK293T cells cotransfected for 48 h with Myc-FBXW7, plus Flag-SHP2, HA-ub, GFP vectors and treated or untreated with MG132. (**f**) Immunoblot analysis of HEK293T cells cotransfected with Myc-FBXW7, HA-ub and GFP, along with vector for Flag-SHP2 (WT), Flag-SHP2 (558A and 562A). (**g**) Immunoblot analysis of the ubiquitination of SHP2 in HEK293T cells cotransfected with Flag-SHP2, HA-ub, along with increasing concentrations (wedge) of vectors for the Myc-FBXW7 constructs and treated with MG132 for 6 h before cell harvest. (**h**) Immunoblot analysis of the ubiquitination of SHP2 in HEK293T cells cotransfected with Flag-SHP2, Myc-FBXW7 along with HA-Ub, the mutant ubiquitin HA-R48 Ub or HA-R63 Ub and treated with MG132 before cell harvest. (**i**) Immunoblot analysis of the K48 ubiquitination of SHP2 in FBXW7[f/f] and Lysm[+]FBXW7[f/f] peritoneal macrophages treated with MG132 and infected with VSV. (**j**) Immunoblot analysis of the ubiquitination of SHP2 in HEK293T cells cotransfected with Myc-FBXW7, HA-ub, along with Flag-SHP2 (WT), Flag-SHP2 (91R) or Flag-SHP2 (178R) expression vector and treated with MG132 before cell harvest. (**k**) Immunoblot analysis of HEK293T cells cotransfected with Myc-FBXW7, HA-ub and GFP, along with vector for Flag-SHP2 (WT), Flag-SHP2 (91R) and treated with MG132 before cell harvest.

were overexpressed and purified (Supplementary Fig. 8d). The *in vitro* interaction experiment showed that FBXW7 directly bounded to SHP2 and the interactions between FBXW7 and RIG-I was mediated by SHP2 (Supplementary Fig. 8e). We also confirmed that SHP2 negatively regulates the RIG-I protein level by comparing RIG-I protein between SHP2-deficient and WT macrophages upon VSV infection through immunoblotting (Fig. 7g). However, there was no significant difference in protein

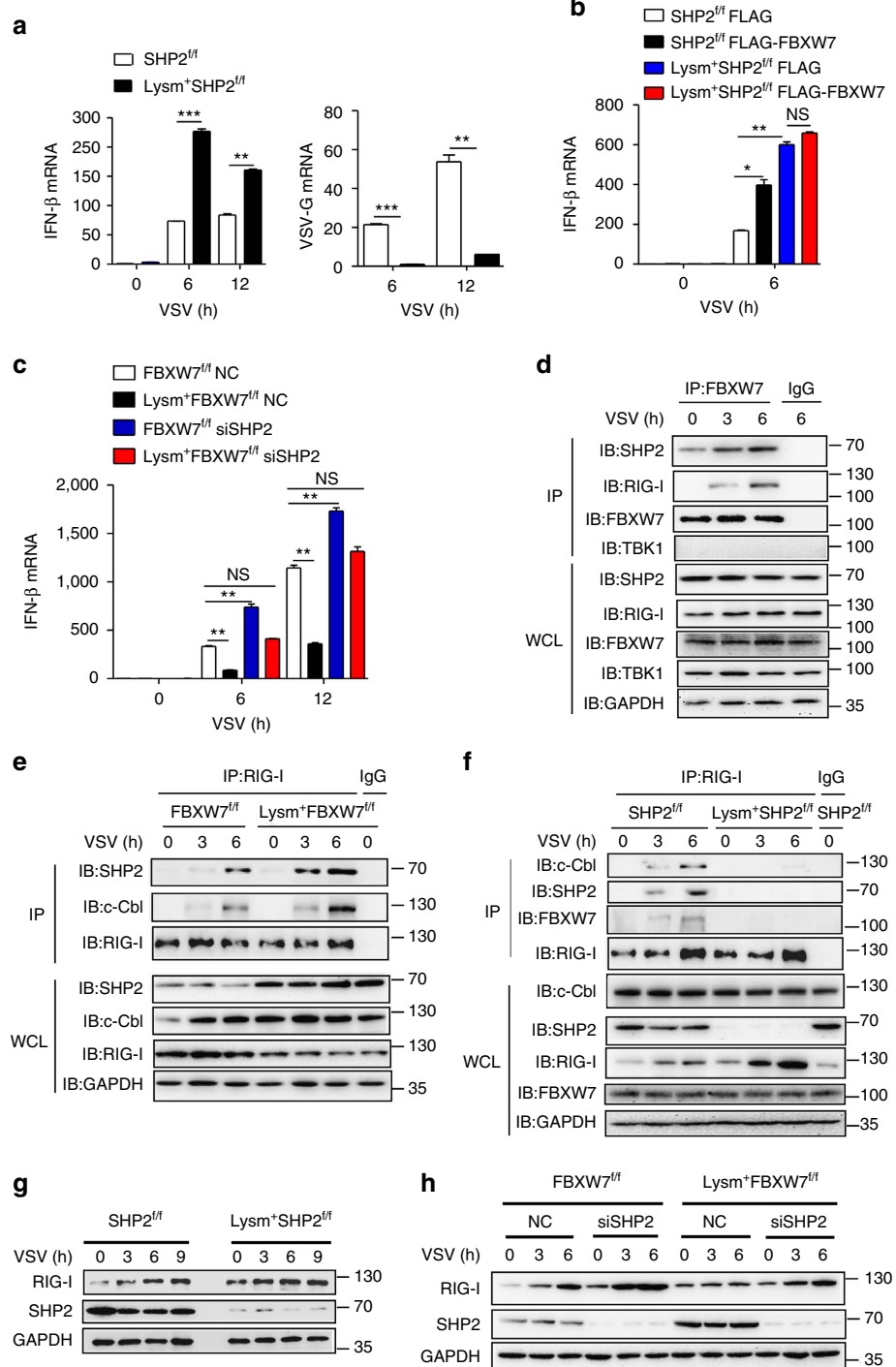

**Figure 7 | The role of FBXW7 stabilizing RIG-I and promoting the production of type I IFN is mediated by SHP2.** (**a**) Quantitative PCR (Q-PCR) analysis of IFN-β and VSV-G mRNA expression in SHP2$^{f/f}$ and Lysm$^+$SHP2$^{f/f}$ peritoneal macrophages infected with VSV. (**b**) Q-PCR analysis of IFN-β mRNA expression in SHP2$^{f/f}$ and Lysm$^+$SHP2$^{f/f}$ peritoneal macrophages transfected for 24 h with Flag-FBXW7 and infected with VSV. (**c**) Q-PCR analysis of IFN-β mRNA in FBXW7$^{f/f}$ and Lysm$^+$FBXW7$^{f/f}$ peritoneal macrophages transfected for 36 h with siSHP2 and infected with VSV. (**d**) Immunoprecipitation and immunoblot analysis of RAW264.7 infected for indicates hours with VSV followed by immunoprecipitation with FBXW7-conjugated magnetic bead or immunoglobulin G (IgG)-conjugated magnetic bead. (**e**) Immunoprecipitation and immunoblot analysis of FBXW7$^{f/f}$ and Lysm$^+$FBXW7$^{f/f}$ peritoneal macrophages infected for indicate hours with VSV followed by immunoprecipitation with RIG-I-conjugated magnetic bead or immunoglobulin G (IgG)-conjugated magnetic bead. The equal amount of RIG-I for different groups was used for western blot analysis. (**f**) Immunoprecipitation and immunoblot analysis of SHP2$^{f/f}$ and Lysm$^+$SHP2$^{f/f}$ peritoneal macrophages infected for indicated hours with VSV followed by immunoprecipitation with RIG-I-conjugated magnetic bead or immunoglobulin G (IgG)-conjugated magnetic bead. (**g**) Immunoblot analysis of RIG-I protein level in lysates of SHP2$^{f/f}$ and Lysm$^+$SHP2$^{f/f}$ macrophages infected with VSV. (**h**) Immunoblot analysis of RIG-I protein in lysates of FBXW7$^{f/f}$ and Lysm$^+$FBXW7$^{f/f}$ macrophages transfected for 36 h with SHP2-siRNA (siSHP2) and infected with VSV. Data are mean ± s.e.m. and are representative of three independent experiments. Student's $t$-test was used for statistical calculation. *$P < 0.05$, **$P < 0.01$ and ***$P < 0.001$.

level of RIG-I between FBXW7[f/f] and Lysm + FBXW7[f/f] macrophages that transfected with siRNA targeting SHP2 (Fig. 7h). All these results demonstrated that the effect of FBXW7 protecting RIG-I from degradation and promoting type-I IFN is dependent on SHP2.

## Discussion

Ubiquitination is a versatile posttranslational modification involved in various cellular functions[41]. Through the past decade, E3 ligase FBXW7 has been reported to play critical roles in tumour initiation and progression, cell proliferation, differentiation and angiogenesis[32]. Although previous researches reported that FBXW7 controlled Merkel cell polyomavirus replication by targeting large T-viral oncoprotein for proteasomal degradation. However, the function of FBXW7 in antiviral immune response is largely unknown. Our study demonstrated the function of FBXW7 in antiviral immunity and the underlying mechanisms. The mice with deletion of FBXW7 in myeloid lineages showed much more sensitivity to RNA virus infection and virus replication was significantly enhanced. In addition, we found that deletion of FBXW7 resulted in reduced production of type-I IFN when cells were infected with RSV, VSV or H1N1 virus, except EMCV. It was implied that the antiviral function of FBXW7 was depending on promoting the production of type-I IFN. Although it was reported that FBXW7α attenuates inflammatory signalling by downregulating C/EBPδ and its target gene Tlr4 (ref. 42), whether FBXW7 regulates inflammatory response induced by other TLR ligands remains unknown. We found Lysm + FBXW7[f/f] macrophages produced less type-I IFN and proinflammatory cytokines when cells stimulated with lipopolysaccharide, poly (I:C) and CPG. Further investigation will be required to identify the function of FBXW7 in antibacterial immunity.

Many cellular proteins have activities that must be controlled, both spatially and temporally, so that appropriate and timely deal could be carried out with cellular stress responses. The FBXW7 gene locus encodes three isoforms (α, β and γ), FBXW7-α, which is located in the nucleoplasm, is detected in all murine tissues. FBXW7-β, which is located in the cytoplasm, is restricted to the brain and testis. FBXW7-γ, which is located in the nucleus, is restricted to heart and skeletal muscles. FBXW7-α is thought to perform most of the FBXW7 functions, although specific roles for the other isoforms have also been described[32]. FBXW7 contains nuclear localization sequences, allowing its localization to the nucleus in cellular resting state. In addition, FBXW7 contains a nuclear export sequence (NES); however, whether FBXW7 could translocate into the cytoplasm and what is the function of FBXW7 nuclear export are unknown. We found FBXW7, refer in particular to FBXW7-α, could translocate from the nucleus into the cytoplasm, when cells were infected with virus. The interaction between FBXW7 and RIG-I was initiated by virus infection; here we demonstrated virus-induced translocation of FBXW7 into the cytoplasm is crucial for RIG-I stabilization, which is critical to recognize viral RNA and trigger its downstream signalling to produce type-I IFN-mediated antiviral immune response. Furthermore, we found deletion of FBXW7 led to the RIG-I instability, which is caused by increased K48-linked polyubiquitination. The E3 ligase RNF125 and c-Cbl were reported to mediate K48-linked ubiquitination of RIG-I and subsequent degradation of RIG-I, which led to decreased viral RNA recognition[22,23]. However, it was observed that FBXW7 could not affect the stability and ubiquitination of MDA5 and MAVS, indicating that FBXW7-promoted antiviral immunity was selectively mediated by RIG-I. Our study first identified FBXW7 as an interacting partner of RIG-I through protecting RIG-I from degradation and positively regulated RIG-I-triggered antiviral immunity.

Our results confirmed that FBXW7 stabilizes RIG-I by targeting SHP2 for its ubiquitination and degradation. SHP2 plays a critical role in regulating innate immunity. SHP2 negatively regulated TLR4- and TLR3-activated IFN-β production partially through inhibiting TBK1-activated signal transduction[43]. TLR4-activated K63-ubiquitination of TRAF6 was affected by SHP2 (ref. 44). SHP2 mediates C-type lectin receptor-induced activation of the kinase Syk and antifungal Th17 response[45]. In addition, we have demonstrated that SHP2 recruits E3 ligase c-Cbl to RIG-I, results in RIG-I degradation and negatively regulates the production of type-I IFN. However, the dephosphorylated function of SHP2 and the posttranslational modification of SHP2 remain largely unknown during virus infection. In this study, we found FBXW7 interacted with SHP2 and mediates the degradation and K48 ubiquitination of SHP2. The interaction between FBXW7 and SHP2 was significantly increased after VSV infection at various time points, which implied that the spatial structure of FBXW7 and SHP2 were changed in VSV-induced signal transduction, resulted in higher affinity of FBXW7 and SHP2. The proper substrate phosphorylation events are required for FBXW7 to recognize and bind its substrates for degradation[38]. Our data demonstrated that the serines at position 558 and 562 of SHP2 were essential for its interaction with FBXW7. It was reported that SHP2 (558S and 562S) phosphorylation was detected by mass spectrometry. However, the kinase that mediates the phosphorylation of SHP2 (558S and 562S) and the role of SHP2 (558S and 562S) phosphorylation in VSV infection still remain to be elucidated. Most of FBXW7 substrates are master transcription factors; thus, FBXW7 regulates diverse pathways with oncogenic potential[32]. Here we found SHP2, a protein tyrosine phosphatase, is the substrate of FBXW7 that has not been reported. We also demonstrated that the interaction between RIG-I and FBXW7 was mediated by SHP2. Although SHP2 could also interact with TBK1, the interaction between SHP2 and RIG-I was significantly increased during VSV infection and FBXW7 predominantly binds to RIG-I along with SHP2 rather than TBK1. Our findings demonstrated that the interaction of FBXW7 and SHP-2 is critical to promote RIG-I signalling pathway to induce sufficient type-I IFN production for protecting against virus.

FBXW7 acts as a tumour suppressor, its gene mutations and reduced expression were reported in many types of human cancers. Here we found that the FBXW7 mRNA expression decreased in PBMCs from 70 paediatric patients infected with RSV when compared with those from 40 healthy children. It indicated that the expression of FBXW7 might have a correlation with the antiviral immunity in humans. Some viruses use evasion strategies to support persistent infection and spread[16,46,47]. It was reported that disruption of RIG-I signalling by the viral NS3/4A protease contributes to the ability of HCV to control innate antiviral defense[48]. Clarifying the molecular mechanisms by which RIG-I expression level and RIG-I downstream signalling pathways are precisely controlled is important for exploring potential targets for therapeutic strategies. Overall, our data have discovered a role of E3 ubiquitination ligase FBXW7 in the positive regulation of RIG-I-triggered antiviral immune response through enhanced IFN production. These findings provide interesting insights into the function of FBXW7 in antiviral immunity and its related clinical significance.

## Methods

**Mice and reagents.** FBXW7[f/f] mice on a C57BL/6J background[49] were obtained from Jackson Laboratories. Lysm-Cre mice C57BL/6J were kindly provided by Dr Ximei Wu, Zhejiang University School of Medicine. Lysm + SHP2

mice C57BL/6J were kindly provided by Dr Yuehai Ke, Zhejiang University School of Medicine. All mice were housed in the University Laboratory Animal Center. All animal experiments were approved by the review committee from Zhejiang University School of Medicine and were in compliance with institutional guidelines. RAW264.7 macrophages (American Type Culture Collection, ATCC, TIB-71) and HEK293T cells (ATCC, CRL-11268) were maintained in DMEM medium, Thioglycolate (BD, 3190383, Merck, 1.08191.0500)-elicited mouse peritoneal macrophages were cultured in RPMI-1640 medium with 10% (vol/vol) FCS. BMDCs and BMDMs were generated from the bone marrow of 6- to 8-week-old mice. Bone marrow cells were collected from the femurs and tibias of mice, and BMDMs were differentiated in RPMI-1640 medium with 10% (vol/vol) FCS and 20 ng ml$^{-1}$ recombinant mouse macrophage colony-stimulating factor (R&D, 416-ML). To generate BMDCs, the bone marrow cells were flushed out of the femurs and tibias, and cultured in RPMI-1640 medium with 10% (vol/vol) FCS, recombinant mouse 10 ng ml$^{-1}$ granulocyte–macrophage colony-stimulating factor (PeproTech, 315-03) and mouse 5 ng ml$^{-1}$ IL-4 (PeproTech, 214-14). Antibodies to the haemagglutinin tag (sc-805, 1:1,000), Myc tag (sc-40; sc-789, 1:1,000), Flag tag (sc-807, 1:1,000), β-actin (sc-130619, 1:1,000), ubiquitin (sc-271289, 1:1,000), CRM1 (sc-74455, 1:200), SHP2 (sc-280, 1:2,000) and SHP2 (C-18AC; sc-280AC, 1:100) were from Santa Cruz, Inc. Antibodies specific for RIG-I (D14G6, 1:500), MDA-5 (5321, 1:2,000), MAVS (4983, 1:2,000), TBK1 (D1B4, 3504, 1:1,000), IRF3 (4962, 1:1,000), IRF3 phosphorylated at Ser396 (4D4G; 4947, 1:5,000), p-JNK (9251, 1:3,000), JNK (9252, 1:1,000), p-p65 (3033, 1:3,000), p65 (8242, 1:1,000), p-p38 (9215, 1:3,000) and p38 (9212, 1:1,000) were from Cell Signaling Technology. Lys48-specific linked polyubiquitin antibody (05-1307, 1:1,000) was from Millipore. Antibodies to the FBXW7 were from Abcam (ab12292, 1:1,000) and Thermo Fisher (40-1500, 1:3,000). MG132 (M8699), CHX (C4859) and Flag-M2 Magnetic Beads (M8823) were from Sigma-Aldrich. ELISA kits for mouse IL-6 and TNF-α were from eBioscience, and IFN-β was from PBL Biomedical Laboratories.

**Plasmid constructs and transfection.** Recombinant vectors encoding mouse FBXW7 (NM_001177773; NM_080428.3) and SHP2 (NM_001109992.1) were created by PCR-based amplification of RAW264.7 complementary DNA, followed by subcloning into the pcDNA3.1 eukaryotic expression vector (Invitrogen) as described. The HA-K48R Ub and HA-K63R Ub were provided by Dr Xu Xiongfei (Second Military Medical University, Shanghai). The sequences for primers are listed in Supplementary Table 1. All constructs were confirmed by DNA sequencing. The plasmids were transfected into HEK293T cells with JetPrime (Polyplus). Primary macrophages and DCs were transfected with siRNA through the use of INTERFERin reagent (Polyplus) according to the standard protocol. FBXW7 siRNA sequences were 5′-ACCTTCTCTGGAGAGAGAAATGC-3′, 5′-GTGTGGAATGCTGAAACTGGAGA-3′ and 5′-CACAAAGCTGGTGTGT GCA-3′ from Life Technology.

**Flow cytometry.** Primary macrophages, BMDMs, BMDCs or HEK293T cells were infected with GFP-VSV for indicated time and analysed by flow cytometry (FACS). For detection of the myeloid cell subsets of FBXW7 KO mice, spleen cells were stained with antibodies specific for mouse CD11b (PE, e-Bioscience, 12-0112), F4/80 (fluorescein isothiocyanate (FITC), e-Bioscience, 11-4801), CD11c (APC, e-Bioscience, 17-0114) and Ly6C (FITC, e-Bioscience, 11-5931-81), and analysed by FACS.

**Mass spectrometry.** HEK293T cells were transfected with Flag-FBXW7 plasmid for 24 h and pretreated with MG132 (25 μM) for 2 h, then infected with VSV for 6 h before collecting. The cell lysates were immunoprecipitated by antibody to Flag epitope tag and mass spectrometry was used to identify FBXW7-interacting proteins.

**Quantitative reverse transcriptase–PCR.** Total RNA was isolated from cells using TRIzol reagent (Takara) according to the manufacturer's directions. Single-strand cDNA was generated from total RNA and reverse transcriptase (Toyobo). The SYBR Green master Rox (Roche) were used for quantitative real-time reverse transcriptase–PCR analysis as described. The sequences for primers are listed in Supplementary Table 2.

**Cytokine release assay.** IL-6, TNF-α and IFN-β cytokines were detected with ELISA kits according to the manufacturer's protocols[50].

**Virus infection.** Primary macrophages or DCs were infected with RSV (subtype A, strain Long, kindly provided by Dr Jing Qian, Zhejiang University School of Medicine, Hangzhou, multiplicity of infection (MOI) = 10), VSV (kindly provided by Dr Huazhang An, Second Military Medical University, Shanghai, MOI = 1), Influenza A virus PR8/A/34 (H1N1, kindly provided by Dr Jing Qian, MOI = 10), EMCV (from ATCC, MOI = 10), herpes simplex virus-1 (KOS strain, kindly provided by Dr Dongli Pan, Zhejiang University School of Medicine, MOI = 10) and VSV-GFP (kindly provided by Dr Zongping Xia, Life Science Institute,

Zhejiang University, MOI = 1) for 12 or 24 h. HEK293T cells were infected with VSV (MOI = 0.1) or VSV-GFP (MOI = 0.1). For mice survival assays, age (6 weeks old)- and sex (male)- matched groups of littermate mice were infected with 1 MLD50 VSV via intraperitoneal injection or infected with 2 MLD50 H1N1 virus by intranasal inoculation.

**Lung histology.** Lungs from control or virus-infected mice were dissected, fixed in 10% phosphate-buffered formalin, embedded into paraffin, sectioned, stained with haematoxylin and eosin solution, and examined by light microscopy for histological changes.

**Immunoprecipitation and immunoblot analysis.** For immunoprecipitation, whole-cell extracts were lysed in IP Lysis Buffer (Pierce, 87785) and a protease inhibitor 'cocktail' (Sigma, P8340). Cell lysates were centrifuged for 15 min at 12,000 g, supernatants were collected and incubated with protein G magnetic beads (Pierce, 88848) together with specific antibodies. After overnight incubation, protein G magnetic beads were washed five times with IP wash buffer. Immunoprecipitates were eluted by SDS–PAGE loading buffer or Elution buffer (Pierce, 88848). For immunoblot analysis, cells were lysed with cell lysis buffer (Cell Signaling Technology, 9803) supplemented with a protease inhibitor 'cocktail' (Sigma, P8340). Protein concentrations in the extracts were measured by BCA assay (Pierce, 23235). Equal amounts of extracts were separated by SDS–PAGE, then transferred onto polyvinylidene fluoride membrane (Millipore, IPVH00010), blocked with 5% dry non-fat milk in Tris-buffered saline (pH 7.4) containing 0.1% Tween-20 and probed with the antibody for immunoblot analysis. The uncropped scans of the western blottings were provided as the Supplementary Figs 9–12 in the Supplementary Information.

**Immunofluorescence staining.** Primary macrophages plated on glass coverslips in six-well plates were infected or uninfected with VSV at 6 h. Cells were fixed with 2% paraformaldehyde for 30 min at 4 °C, permeabilized using 0.1% Triton X-100, blocked with 1% BSA in PBS for 1 h and stained with rabbit anti-FBXW7 or anti-Flag antibody and mouse anti-CRM1, anti-SHP2 or anti-Myc antibody, followed stained FITC-labelled anti-rabbit secondary antibody and Texas Red-labelled anti-rabbit secondary antibody, to detect co-localization of FBXW7 and CRM1/SHP2/RIG-I. The nuclei were stained with DAPI (4, 6-diamidino-2-phenylindole; Sigma). The co-localization was finally detected with a Leica TCS SP2 confocal laser microscope.

**Human peripheral blood samples.** A total of 70 peripheral blood samples were collected from children with bronchiolitis hospitalized in Children's Hospital, Zhejiang University School of Medicine, China, from 1 April 2010 to 31 December 2014, and 40 healthy children as a control group. The diagnosis and severity evaluation of bronchiolitis were according to the Expert consensus document on diagnosis and treatment of bronchiolitis (2014). The RSV infection was confirmed by RSV antigen tests of nasopharyngeal aspirates. Furthermore, other microbiologic tests were done to exclude other respiratory tract infections and tuberculosis, including protein-purified derivative test, blood cultures, nasopharyngeal aspiratefor other common respiratory tract virus antigens (influenza virus, adenovirus, metapneumovirus and parainfluenza virus), and serology for *Mycoplasma pneumonia*, *Chlamydia pneumoniae* and *Legionella pneumophila*. No other pathogens were found by these tests. Peripheral blood samples were obtained from the patients on admission. The study was approved by the ethics committee of the Children's Hospital, Zhejiang University School of Medicine. Written informed consent was obtained from at least one guardian of each patient before enrollment. The data from patients were analysed anonymously.

**Statistical analysis.** The significance of difference between groups was determined by two-tailed Student's $t$-test and two-way analysis of variance test. For mouse survival study, Kaplan–Meier survival curves were generated and analysed for statistical significance with GraphPad Prism 4.0. $P$-values of $< 0.05$ were considered statistically significant.

**Data availability.** All relevant data are available from the authors upon request.

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

## Acknowledgements

We thank Dr Wu Ximei for providing the Lysm-Cre mice, Dr Ke Yuehai for Lysm$^+$ SHP2$^{f/f}$ mice, Dr Wang Xiaojian for TBK1, MAVS and RIG-I plasmids, and Dr Xia Zongping for VSV-GFP. This work was supported by the National Natural Science Foundation of China (81230074, 81571524 and 81202295) and the National Key Basic Research Program of China (2013CB530500).

## Author contributions

Q.W., Y.L. and X.C. designed and supervised the research. Y.S., Y.L., L.L., Z.C., J.H., Y.Z., Y.X., Y.X., S.C., P.D., F.D. and L.C. performed research. M.Z., X.W. and P.X. contributed new reagents/analytical tools. Q.W., Y.L. and Y.S. analysed data. Q.W., Y.S. and Y.L. wrote the paper.

## Additional information

**Competing financial interests:** The authors declare no competing financial interests.

