## [Peer Review File · Nature Communications]

Reviewers' comments:

Reviewer #1 (Remarks to the Author):

In this manuscript, Song et al. identified FBXW7 as a new positive regulator for RIG-I-triggered type I IFN signaling pathway. The authors observed the phenotype of a reduction of FBXW7 mRNA in PBMCs from 70 cases of children infected with RSV relatively to the healthy control and the impaired antiviral immunity in FBXW7 deficient mice or macrophages. Mechanistically, the authors found, upon VSV infection, FBXW7 translocated from the nucleus into the cytoplasm through CRM1, interacted with RIG-I and SHP2, and ubiquitinated and degraded SHP2 through its E3 ligase activity, thereby preventing the RIG-I degradation by SHP2/c-Cbl complex.

The FBXW7-containing SCF complex regulated the ubiquitination and degradation of multiple substrates through E3 ligase activity, as described in the section of introduction in this manuscript. The findings provided in this manuscript that FBXW7 interacted with SHP2 to stabilize RIG-I and promote type I IFN production could expand the diversity of regulatory mechanisms for FBXW7 ligase.

Major concerns:

1. FBXW7 interacted with RIG-I and SHP-2. FBXW7 seems to regulate K48 ubiquitin conjugation on both RIG-I and SHP2 in a viral infection-specific manner. As authors insisted, the K48-ubiquitination of SHP2 seems to be catalyzed by FBXW7, but how and why K48-ubiquitinated RIG-I was not degraded by the same E3 ligase in a same binding complex? This is still unclear.

2. Previous studies showed that upon nucleoprotein export to cytoplasm, CRM1 protein also translocated together with targeted protein (Fornerod et al., Cell. 1997). However, the localization of CRM1 did not change but still stayed in the nucleus upon viral infection in Fig. 4A. Is this different mechanism from previously known function? The use of the selective inhibitor of CRM1/XPO1 could be considered. In addition, author should insert the reference when originally introducing CRM1 in the section of results.

3. In Fig. 1A, the authors insisted that FBXW7 mRNA level between healthy and RSV infected individuals are significantly different. But it seems there was a marginal or slight difference. Is this difference statistically meaningful? On the line of 141, the authors said "IFN-beta mRNA expression was extremely attenuated", however the data showed about 50% reduction. This may lead to over-interpretation.

4. In Fig. 6K, based on the authors' hypothesis, over expression of FBXW7 should reduce SHP2 level through K48 Ubiquitin-mediated degradation. However, the reduction of Flag-SHP2 was not correlated with the level of Myc-FBXW7. This data is not convincing.

5. In Fig S2 and Fig S4, the authors tested the effect of FBXW7 on IFN signaling with poly(I:C) transfection. It is well known that long poly(I:C) selectively activates MDA5 while short poly I:C induces IFN signaling through RIG-I. Which poly(I:C) did authors use for this experiment? Did FBXW7 also affect MDA5's stability or affect only RIG-I? The authors should

test if FBXW7 also regulated MDA5 stability by using EMCV or TMEV.

6. In Fig. 5D and S6B, the localization of FBXW7 seems already be in the cytoplasm without viral infection. The authors should show the nucleus staining. Also, all the immunofluorescence data in this manuscript (Fig 3F, Fig 4A, 4C, 4E, Fig 5D, Fig S5A, S5B, Fig. S6B) should be quantified by multiple images per sample rather than just showing one single cell image.

7. The authors detected a significant decrease of FBXW7 mRNA in PBMCs from 70 cases of children infected with RSV compared to healthy children. However, all the animal studies including mouse survival was done using VSV and partially influenza H1N1 subtype. Given that authors firstly detected this phenomenon from RSV infected children, it is relevant to do mouse survival study using RSV or influenza virus.

Minor concerns:

1. In Fig. 6E, there was no effect of Myc-FBXW7 on the SHP2 reduction. Also the effect of MG132 was very small. This result should be improved using quantitative way. In addition, why the authors used GFP vector for several experiments rather than checking beta-actin?

2. In Fig. 7E and 7F, there was no input control (WCL). The authors should add it. At FIG.7E, if the authors' hypothesis was true, FBXW7 KO cell should show the lower RIG-I protein level. Thus, WCL input result is required to be shown.

3. In the line of 155 on page 5, authors described ' 8×10^7 pfu/g viruses were injected via the tail vein' which is IV injection. However, on the line of 500 in the methods section, authors described that VSV were infected via intraperitoneal injection. Authors should clarify the route of injection.

4. Author made mistake to mark Figure in the manuscript. In the line 182, 191, 193, 233, 243 and 245, it should be Figure S3B, Figure S3G, Figure S3H, Figure S3I, Figure S5A, Figure S5B and Figure S5C. The mistakes like this should not have occurred.

5. In Fig. 3G and 6B, the authors should make the graph legend of X and Y axis clear.

6. In Fig. 6H, there should be an explanation of usage of the mutant ubiquitin K48R and K63R in the manuscript.

7. In Fig. 7D, the data quality should be improved.

8. We do not see the meaning for asterisk from Figure 5F in the figure legend or the results section. Author should add it to make clear.

9. On the line of 497 in the Methods of virus Infection section, H1NI should be H1N1.

10. In the Methods of virus infection section, authors should describe in more detail about strain of viruses used in this study and how much MLD50 of viruses used in mouse experiments.

Reviewer #2 (Remarks to the Author):

In this manuscript, Song et al. demonstrated an important function of the E3 ubiquitin ligase FBXW7 in innate antiviral signaling. After RNA virus infection, they found FBXW7 translocated from the nucleus into the cytoplasm, where FBXW7 binds to and promotes SHP2 ubiquitination and degradation, thereby leading to stabilization of the dsRNA receptor RIG-I, which could be degraded by SHP2/c-Cbl complex. Thus, *Lysm+FBXW7^{f/f}* mice showed impaired antiviral immunity compared to WT mice. These findings identified a novel function of FBXW7 in antiviral immunity and delineated a novel mechanism for the regulation of RIG-I stability.

Major concerns:

The authors found that RIG-I/SHP2/FBXW7 formed a complex after virus infection and claimed that the interaction between RIG-I and FBXW7 was mediated by SHP2. But, there is no strong evidence to support this claim. The direct interaction among RIG-I, FBXW7, and SHP2 should be examined with recombinant proteins.

SHP2 negatively regulated TLR4- and TLR3-activated IFN- β production through inhibiting TBK1-activated signal transduction. TBK1 is also an essential kinase in RIG-I-mediated IFN- β signaling. Is there a possibility that FBXW7 regulates RIG-I signaling through SHP2/TBK1 except SHP2/RIG-I?

MAVS is key adaptor in the RIG-I mediated signaling? Is FBXW7 regulates MAVS activation such as ubiquitination? These experiments should be very important controls to confirm FBXW7 specifically targets RIG-I.

The authors found that RNA virus-induced IFN- β signaling was regulated by FBXW7 through stabilization of RIG-I. RNA virus could also be detected by TLR3, thus, what is the function of FBXW7 in TLR3-mediated IFN- β signaling? How about the direct stimulation of the cells with poly(I:C), which will be detected by TLR3?

The authors found DNA virus HSV-1-induced IFN- β was not impaired in KO mice (Fig. S2E). These phenomena need more experiments to confirm, such as the experiments using macrophages and DCs, the survival of HSV-1 challenged KO mice.

Fig.S1 D and E, the mRNA of FBXW7 was greatly decreased in KO cells (D), why the expression of FBXW7 was increased in KO cells as measured by WB?

Minor concerns:

Fig. 4C and E, why the cellular location of Flag-FBXW7 showed different patterns in the same HEK293 cells especially for the mutants 4A and F box deletion? Whether could these experiments be repeated?

Fig.5A, why the protein level of Flag-FBXW7 was increased in the IP? The increased interaction between SHP2 and Flag-FBXW7 was due to increased protein level of Flag-FBXW7? The experiments should be repeated.

Fig.6E, the protein level of Flag-SHP2 in lane 3 was not decreased compared to that in lane 2 and 4, while, the data in Fig.6D showed a clear decrease in the presence of FBXW7. The experiment should be repeated.

Fig. 6G, the ubiquitination of Flag-SHP2 was increased in the presence of FBXW7 (lane 4 and 5). Why the protein level of Flag-SHP2 was not decreased in the presence of FBXW7 (lane 4 and 5)? Same problems in Fig.6H and J, Fig.S7E,F,G.

The important input controls were missing in several Figures. Fig.4F, Fig.6H, I, J, Fig.7E, F et al.

Several blots in Fig. 7 (D,H) are in low quality. These experiments should be repeated.

We thank the reviewers for their excellent comments and suggestions, those comments are all valuable and very helpful for revising and improving our paper. We do our best to answer their questions, the point-by-point responses to the comments are attached below.

Point-by-Point Responses to Comments from the Reviewers

Reviewers' comments are *italicized* and our responses are in normal text.

Reviewer #1:

In this manuscript, Song et al. identified FBXW7 as a new positive regulator for RIG-I-triggered type I IFN signaling pathway. The authors observed the phenotype of a reduction of FBXW7 mRNA in PBMCs from 70 cases of children infected with RSV relatively to the healthy control and the impaired antiviral immunity in FBXW7 deficient mice or macrophages. Mechanistically, the authors found, upon VSV infection, FBXW7 translocated from the nucleus into the cytoplasm through CRM1, interacted with RIG-I and SHP2, and ubiquitinated and degraded SHP2 through its E3 ligase activity, thereby preventing the RIG-I degradation by SHP2/c-Cbl complex.

The FBXW7-containing SCF complex regulated the ubiquitination and degradation of multiple substrates through E3 ligase activity, as described in the section of introduction in this manuscript. The findings provided in this manuscript that FBXW7 interacted with SHP2 to stabilize RIG-I and promote type I IFN production could expand the diversity of regulatory mechanisms for FBXW7 ligase.

Major concerns:

C1

FBXW7 interacted with RIG-I and SHP-2. FBXW7 seems to regulate K48 ubiquitin conjugation on both RIG-I and SHP2 in a viral infection-specific manner. As authors insisted, the K48-ubiquitination of SHP2 seems to be catalyzed by FBXW7, but how and why K48-ubiquitinated RIG-I was not degraded by the same E3 ligase in a same binding complex? This is still unclear.

R1

We thank the reviewer for raising this important issue. We are sorry for having not explained very clearly. In our study it was firstly expected that K48-ubiquitination and degradation of substrates catalyzed by E3 ligase FBXW7 might be decreased in Lysm⁺FBXW7^{fl/fl} macrophages during VSV infection. Surprisingly the results showed that K48-ubiquitination and degradation of RIG-I were not downregulated but enhanced in Lysm⁺FBXW7^{fl/fl} macrophages (Figure 3b, 4f), so RIG-I was not the direct substrate of FBXW7. It made us to further explore the mechanisms involved in the regulation of RIG-I by FBXW7. We then found the K48-ubiquitination and degradation of SHP2 were

downregulated in $\text{Lysm}^+\text{FBXW7}^{\text{ff}}$ macrophages during VSV infection (Figure 6a, 6i). The interaction between RIG-I and FBXW7 was not observed in VSV infected $\text{Lysm}^+\text{SHP2}^{\text{ff}}$ macrophages, indicating that the interaction between FBXW7 and RIG-I is mediated by SHP2 (Figure 7f). Also, we demonstrated that FBXW7 directly bound to SHP2 and the interaction between FBXW7 and RIG-I was mediated by SHP2 (Figure S8e). Below is the working model explaining how FBXW7 promotes the RIG-I stabilization. FBXW7 targets SHP2 and mediates the degradation and ubiquitination of SHP2, thus disrupting the SHP2/c-Cbl complex which mediated RIG-I degradation.

Working model: when innate cells were stimulated with RNA virus, E3 ubiquitin ligase FBXW7 translocates from the nucleus into the cytoplasm to stabilize RIG-I by targeting SHP2 for ubiquitination, so FBXW7 is crucial for the stability of RIG-I and RIG-I-triggered type I IFN production.

C2

Previous studies showed that upon nucleoprotein export to cytoplasm, CRM1 protein also translocated together with targeted protein (Fornerod et al., Cell. 1997). However, the localization of CRM1 did not change but still stayed in the nucleus upon viral infection in Fig. 4A. Is this different mechanism from previously known function? The use of the selective inhibitor of CRM1/XPO1 could be considered. In addition, author should insert the reference when originally introducing CRM1 in the section of results.

R2

We thank for the reviewer's important suggestion, and we used Leptomycin B (LMB) which binds to the active site (CRM1 cysteine residue 528) and prevents CRM1 binding to the cargo protein, the result showed that the translocation of FBXW7 from nucleus to cytoplasm was blocked (Figure S5b). Although the localization of CRM1 was almost in the nucleus, we observed CRM1 could also translocated into cytoplasm along with FBXW7 in

bone marrow derived macrophages (BMDM) infected with VSV for 6 hours (Figure 4a). However, the phenomenon that CRM1 translocated into cytoplasm along with FBXW7 was not obvious in HEK293 T cells (Figure S5a), as HEK293 T cells are less sensitive to VSV infection than BMDM. We have inserted the reference when originally introducing CRM1 in the section of results.

C3

In Fig. 1A, the authors insisted that FBXW7 mRNA level between healthy and RSV infected individuals are significantly different. But it seems there was a marginal or slight difference. Is this difference statistically meaningful? On the line of 141, the authors said "IFN-beta mRNA expression was extremely attenuated", however the data showed about 50% reduction. This may lead to over-interpretation.

R3

We thank for the reviewer's important comments. Our data showed that FBXW7 mRNA level between healthy and RSV infected individuals are different, though the difference was not so dramatic, but this difference was statistically meaningful. So we showed this result in the manuscript (Figure 1a). The reviewer's comments are very helpful, we still need more clinical samples to further investigate the difference of FBXW7 mRNA level between healthy and RSV infected individuals.

We really thank for the reviewer's suggestion. We are sorry that the description "IFN-beta mRNA expression was extremely attenuated" (on the line 141) is not very appropriate, so we deleted "extremely". And we repeated this experiment and observed that the IFN- β mRNA expression was significantly attenuated in siRNA-FBXW7 transfected THP-1 cells after infected with RSV (Figure 1b).

C4

In Fig. 6K, based on the authors' hypothesis, over expression of FBXW7 should reduce SHP2 level through K48 Ubiquitin-mediated degradation. However, the reduction of Flag-SHP2 was not correlated with the level of Myc-FBXW7. This data is not convincing.

R4

We are sorry for this confusing result. Since the reference GFP was not very equal in different samples, so the result seemed that reduction of Flag-SHP2 was not correlated with the level of Myc-FBXW7 in the previous figure 6k. We now repeated this experiment and added the new results showing the reduction of Flag-SHP2 was along with increased expression of FBXW7, while the SHP2 (K91R) mutant completely blocked the degradation of SHP2 (Figure 6k).

C5

In Fig S2 and Fig S4, the authors tested the effect of FBXW7 on IFN signaling with poly(I:C) transfection. It is well known that long poly(I:C) selectively activates MDA5 while short poly I:C induces IFN signaling through RIG-I. Which poly(I:C) did authors use for this experiment? Did FBXW7 also affect MDA5's stability or affect only RIG-I? The authors should test if FBXW7 also regulated MDA5 stability by using EMCV or TMEV.

R5

We thank for the reviewer's important suggestion. The low molecular weight (LMW) poly (I:C), with an average of 0.2~1kb, that specifically recognized by RIG-I, while high molecular weight (HMW), with an average of 1.5~8kb, recognized both by RIG-I and MDA5, we used low molecular weight (LMW) poly (I:C) to transfect peritoneal macrophages to trigger the IFN signaling.

MDA5 specifically recognizes picornaviruses, such as encephalomyocarditis virus (EMCV). we detected the protein level of MDA5 between FBXW7^{fl/fl} and Lysm⁺FBXW7^{fl/fl} macrophages infected with EMCV (Figure S4d). There was no significant difference in MDA5 protein level between FBXW7 WT and KO macrophages. The same phenomenon was observed in macrophages transfected with HMW poly (I:C) which could be recognized by MDA5 (Figure S4e). And the half-life of MDA5 protein showed no difference between FBXW7 WT and KO macrophages treated with cycloheximide during VSV infection (Figure S4f, g). FBXW7 deficient macrophages expressed almost the same levels of IFN- β and IFN- α 4 mRNA compared with FBXW7 WT macrophages upon infection with EMCV. These data indicated that FBXW7 specifically interacted with RIG-I and protected RIG-I from degradation, thus promoting RIG-I signaling pathway.

C6

In Fig. 5D and S6B, the localization of FBXW7 seems already be in the cytoplasm without viral infection. The authors should show the nucleus staining. Also, all the immunofluorescence data in this manuscript (Fig 3F, Fig 4A, 4C, 4E, Fig 5D, Fig S5A, S5B, Fig. S6B) should be quantified by multiple images per sample rather than just showing one single cell image.

R6

We thank the reviewer for this important suggestion. In the revised manuscript, we showed the *nucleus staining* and immunofluorescence images with multiple cells per sample (Fig 3f, Fig 4a, 4c, 4e, Fig 5d, Fig S5a, S5b, S5c and Fig. S6b). Also we repeated the experiment and observed the co-localization of FBXW7 and SHP2 in peritoneal macrophages (Figure 5d, S6d).

C7

The authors detected a significant decrease of FBXW7 mRNA in PBMCs from 70 cases of children infected with RSV compared to healthy children. However, all the animal studies including mouse survival was done using VSV and partially influenza H1N1 subtype. Given that authors firstly detected this phenomenon from RSV infected children, it is relevant to do mouse survival study using RSV or influenza virus.

R7

We thank for the reviewer's important suggestion. We designed and utilized RSV infected mice from the beginning. We tried to infected mice with RSV to compare the survival between WT and KO mice, but the mice failed to die, due to the low virulence of RSV strain. We are very sorry it was really hard to complete this mouse survival experiment. So, we had to choose VSV and H1N1 to infect mice. In the revised manuscript, we added the results of the survival of the mice infected by H1N1, Lysm⁺FBXW7^{fl/fl} mice showed higher sensitivity to H1N1 virus infection (Figure S2a).

Minor concerns:

C1

In Fig. 6E, there was no effect of Myc-FBXW7 on the SHP2 reduction. Also the effect of MG132 was very small. This result should be improved using quantitative way. In addition, why the authors used GFP vector for several experiments rather than checking beta-actin?

R1

We thank the reviewer's important advice and we have repeated the experiments and the new results were showed in Figure 6e.

As Myc-FBXW7 and Flag-SHP2 plasmids were transfected into HEK 293T cells, so GFP served as reference to exclude the deviation of transfection efficiency (references of other literatures was listed below).

References:

Inuzuka H, Shaik S, Onoyama I, Gao D, Tseng A, Maser RS, et al. SCF(FBW7) regulates cellular apoptosis by targeting MCL1 for ubiquitylation and destruction. Nature. 2011, 471(7336): 104-109

Wang R, Wang Y, Liu N, Ren C, Jiang C, Zhang K, Yu S, Chen Y, Tang H, Deng Q, Fu C, Wang Y, Li R, Liu M, Pan W, Wang P. FBW7 regulates endothelial functions by targeting KLF2 for ubiquitination and degradation. Cell Res. 2013, 23(6):803-19

C2

In Fig. 7E and 7F, there was no input control (WCL). The authors should add it. At Fig.7E, if the authors' hypothesis was true, FBXW7 KO cell should show the lower RIG-I protein level. Thus, WCL input result is required to be shown.

R2

We thank the reviewer's important advice and we have added these input controls in the revised manuscript. The results of WCL in Figure 7e showed the RIG-I protein level was lower in FBXW7 KO macrophages, and the SHP2 protein level was higher in FBXW7 KO macrophages. Also the results of WCL in Figure 7f displayed the RIG-I protein level was higher in SHP2 KO macrophages.

C3

In the line of 155 on page 5, authors described '8x10⁷ pfu/g viruses were injected via the tail vein' which is IV injection. However, on the line of 500 in the methods section, authors described that VSV were infected via intraperitoneal injection. Authors should clarify the route of injection.

R3

We are sorry for this mistake. We have corrected this mistake in the methods section. The route of injection was clearly indicated in the figure legends.

C4

Author made mistake to mark Figure in the manuscript. In the line 182, 191, 193, 233, 243 and 245, it should be Figure S3B, Figure S3G, Figure S3H, Figure S3I, Figure S5A, Figure S5B and Figure S5C. The mistakes like this should not have occurred.

R4

We are so sorry and have corrected this mistake in the revised manuscript.

C5

In Fig. 3G and 6B, the authors should make the graph legend of X and Y axis clear.

R5

We thank for the reviewer's suggestion. We have indicated the names of X and Y axis in the revised Fig. 3e and 6b.

C6

In Fig. 6H, there should be an explanation of usage of the mutant ubiquitin K48R and K63R in the manuscript.

R6

We thank for the reviewer's important suggestion. We have explained the usage of the mutant ubiquitin K48R and K63R in the revised manuscript.

C7

In Fig. 7D, the data quality should be improved.

R7

We really thank for the reviewer's important suggestion. We have repeated the experiment of Fig. 7d and showed the new results in revised manuscript.

C8

We do not see the meaning for asterisk from Figure 5F in the figure legend or the results section. Author should add it to make clear.

R8

We are so sorry and have made it clear in in the figure legend of Figure 5f. * represents IgG heavy chain.

C9

On the line of 497 in the Methods of virus Infection section, H1NI should be H1N1.

R9

We are so sorry and have corrected this mistake.

C10

In the Methods of virus infection section, authors should describe in more detail about strain of viruses used in this study and how much MLD50 of viruses used in mouse experiments.

R10

We thank for the reviewer's suggestion. We have described more details about strain of viruses used in this study and how much MLD50 of viruses used in mouse experiments in the section of Methods of the revised manuscript.

Reviewer #2

In this manuscript, Song et al. demonstrated an important function of the E3 ubiquitin ligase FBXW7 in innate antiviral signaling. After RNA virus infection, they found FBXW7 translocated from the nucleus into the cytoplasm, where FBXW7 binds to and promotes SHP2 ubiquitination and degradation, thereby leading to stabilization of the dsRNA receptor RIG-I, which could be degraded by SHP2/c-Cbl complex. Thus, Lysm+FBXW7^{fl/fl} mice showed impaired antiviral immunity compared to WT mice. These findings identified a novel function of FBXW7 in antiviral immunity and delineated a novel mechanism for the regulation of RIG-I stability.

Major concerns:

C1

The authors found that RIG-I/SHP2/FBXW7 formed a complex after virus infection and claimed that the interaction between RIG-I and FBXW7 was mediated by SHP2. But, there is no strong evidence to support this claim. The direct interaction among RIG-I, FBXW7, and SHP2 should be examined with recombinant proteins.

R1

We thank for the reviewer's important suggestion. We added this experiment with recombinant proteins to provide the evidence to support the statement "the interaction between RIG-I and FBXW7 was mediated by SHP2". As the interaction between FBXW7 and SHP2 or RIG-I was significantly increased during VSV infection, we overexpressed and purified Flag-FBXW7, Myc-RIG-I, HA-SHP2 tag proteins in HEK293T cells which were infected with VSV for 6 hours before harvest (Figure S8d), then the *in vitro* interaction experiment showed that FBXW7 directly bound to SHP2 and the interaction between FBXW7 and RIG-I was mediated by SHP2 (Figure S8e).

C2

SHP2 negatively regulated TLR4- and TLR3-activated IFN- β production through inhibiting TBK1-activated signal transduction. TBK1 is also an essential kinase in RIG-I-mediated IFN- β signaling. Is there a possibility that FBXW7 regulates RIG-I signaling through SHP2/TBK1 except SHP2/RIG-I?

R2

We thank for the reviewer's important question. We performed these experiments and the results were added in the revised manuscript. The endogenous FBXW7/RIG-I/SHP2 complex was detected in VSV-infected macrophages, while the interaction between FBXW7 and TBK1 was not detected (Figure 7d). It indicated that the interaction between SHP2 and RIG-I was significantly increased during VSV infection, which will lead to

FBXW7 binding to RIG-I along with SHP2 rather than binding to TBK1 (Figure S8c). So our findings confirmed FBXW7 specifically interacted with RIG-I.

C3

MAVS is key adaptor in the RIG-I mediated signaling? Is FBXW7 regulates MAVS activation such as ubiquitination? These experiments should be very important controls to confirm FBXW7 specifically targets RIG-I.

R3

We thank for the reviewer for raising this important issue. We also detected the protein level of MAVS between FBXW7 WT and KO macrophages. There is no significant difference in MAVS protein level between FBXW7 WT and KO macrophages (Figure S4c). And the K48-linked polyubiquitination of MAVS showed no significant difference between FBXW7 WT and KO macrophages (Figure S5h). These results indicate that FBXW7 could not affect the activation of MAVS.

C4

The authors found that RNA virus-induced IFN- β signaling was regulated by FBXW7 through stabilization of RIG-I. RNA virus could also be detected by TLR3, thus, what is the function of FBXW7 in TLR3-mediated IFN- β signaling? How about the direct stimulation of the cells with poly(I:C), which will be detected by TLR3?

R4

We thank for the reviewer's important suggestion. The result was showed below. Lysm⁺FBXW7^{fl/fl} macrophages produced less type I interferon and proinflammatory cytokines when cells stimulated with poly (I:C) (see below figure A). However, the localization of FBXW7 did not change and still stayed in the nucleus upon poly (I:C) stimulation (see below figure B). Further investigation will be required to identify the function and mechanisms of FBXW7 in anti-bacterial immunity. So we did not include this result in the manuscript.

(A) Q-PCR analysis of IFN- β , TNF- α mRNA expression in FBXW7^{fl/fl} and Lysm⁺FBXW7^{fl/fl} peritoneal macrophages stimulated with poly (I:C). (B) Confocal microscopy imaging of BMDM stimulated with poly (I:C) for indicated hours and labeled with antibodies to the appropriate protein. **P<0.01.

C5

The authors found DNA virus HSV-1-induced IFN- β was not impaired in KO mice (Fig. S2E). These phenomena need more experiments to confirm, such as the experiments using macrophages and DCs, the survival of HSV-1 challenged KO mice.

R5

We thank for the reviewer's suggestion. Although there was no significant difference in IFN- β production between FBXW7 WT and KO mice challenged with HSV-1 (Figure S2f). However, we found that the production of type I interferon in macrophages or BMDM and the survival of HSV-1 challenged FBXW7 KO mice showed different effects (see below figure A-D). We will investigate the function and the mechanisms of FBXW7 in response to HSV-1 in the future study. So we did not include this result in the manuscript.

(A-C) Q-PCR analysis of IFN- β mRNA in FBXW7^{fl/fl} and Lysm⁺FBXW7^{fl/fl} peritoneal macrophages (A), BMDM (B), BMDC(C) infected with HSV-1 for indicated time. (D) Survival of FBXW7^{fl/fl} and Lysm⁺FBXW7^{fl/fl} mice infected HSV-1. FB WT and KO mice were infected by 4×10^8 pfu HSV-1 with intraperitoneal injection. Data are mean \pm s.e.m and are representative of three independent experiments. *P < 0.05; **P < 0.01.

C6

Fig.S1 D and E, the mRNA of FBXW7 was greatly decreased in KO cells (D), why the expression of FBXW7 was increased in KO cells as measured by WB?

R6

We are sorry for no having explained very clearly. Because FBXW7 KO macrophages lost the F-box domain of FBXW7 (not the whole deletion), the WB result showed that the molecule weight of FBXW7 was smaller in FBXW7 KO cells (figure S1e). Sorry, due to the loading control in different samples (in figure S1e) was not very equal, the protein level of

the β -actin in FBXW7 KO cells was higher, so it seems that FBXW7 KO cells had the higher protein level of FBXW7. In the qRT-PCR analysis, the primer includes the sequence of F-box domain of FBXW7 and it displayed lower mRNA of FBXW7 in KO cells (figure S1d).

Minor concerns:

C1

Fig. 4C and E, why the cellular location of Flag-FBXW7 showed different patterns in the same HEK293 cells especially for the mutants 4A and F box deletion? Whether could these experiments be repeated?

R1

We thank for the reviewer's question. We have repeated these experiments and showed the new results in the revised manuscript (figure 4c and 4e).

C2

Fig.5A, why the protein level of Flag-FBXW7 was increased in the IP? The increased interaction between SHP2 and Flag-FBXW7 was due to increased protein level of Flag-FBXW7? The experiments should be repeated.

R2

We thank for the reviewer's question and sorry for low quality result. We have repeated these experiments and showed the new results in the revised manuscript (figure 5a).

C3

Fig.6E, the protein level of Flag-SHP2 in lane 3 was not decreased compared to that in lane 2 and 4, while, the data in Fig.6D showed a clear decrease in the presence of FBXW7. The experiment should be repeated.

R3

We thank for the reviewer's suggestion. We have repeated these experiments and showed the new results in the revised manuscript (figure 6e).

C4

Fig. 6G, the ubiquitination of Flag-SHP2 was increased in the presence of FBXW7 (lane 4 and 5). Why the protein level of Flag-SHP2 was not decreased in the presence of FBXW7 (lane 4 and 5)? Same problems in Fig.6H and J, Fig.S7E, F, G.

R4

We are so sorry for not explained clearly. In the experiments about FBXW7 mediated ubiquitination of SHP2, cells were all treated with MG132 which inhibited the degradation mediated by ubiquitin-proteasome system. So the protein level of Flag-SHP2 was not

decreased in the presence of FBXW7. This procedure of MG132 treatment was mentioned in the figure legends.

C5

The important input controls were missing in several Figures. Fig.4F, Fig.6H, I, J, Fig.7E, F et al.

R5

We thank for the reviewer's suggestion. We have added all the input controls and showed them in the revised manuscript.

C6

Several blots in Fig. 7 (D, H) are in low quality. These experiments should be repeated.

R6

We thank for the reviewer's suggestion. We are really sorry for these low quality results. We have repeated these experiments and showed them (figure 7d,h) in the revised manuscript.

REVIEWERS' COMMENTS:

Reviewer #2 (Remarks to the Author):

The authors addressed most of my concerns, the manuscript was much approved.